# Can Muon Fine-tune Adam-Pretrained Models?

Xingyu Qu [* 1]   Peigeng Huang [* 2]   Samuel Horvath [1]

## Abstract

Muon has emerged as an efficient alternative to Adam for pretraining, yet remains underused for fine-tuning. A key obstacle is that most open models are pretrained with Adam, and naively switching to Muon for fine-tuning leads to degraded performance due to an optimizer mismatch. We investigate this mismatch through controlled experiments and relate it to the distinct implicit biases of Adam and Muon. We provide evidence that the mismatch disrupts pretrained knowledge, and that this disruption scales with update strength. This leads us to hypothesize that constraining updates should mitigate the mismatch. We validate this with LoRA: across language and vision tasks, LoRA reduces the performance gap between Adam and Muon observed under full fine-tuning. Studies on LoRA rank, catastrophic forgetting, and LoRA variants further confirm that mismatch severity correlates with update strength. These results shed light on how optimizer mismatch affects fine-tuning and how it can be mitigated. Our code is available here.

## 1. Introduction

Muon (Jordan et al., 2024) (**M**oment**U**m **O**rthogonalized by **N**ewton-Schulz) has emerged as a promising alternative to Adam (Kingma & Ba, 2015; Loshchilov & Hutter, 2019) for large language model (LLM) pretraining. It orthogonalizes the momentum matrix before each update, achieving approximately $2\times$ compute efficiency over Adam (Liu et al., 2025; Shah et al., 2025) while requiring less memory by eliminating the second moment. Notably, it has been successfully adopted for training state-of-the-art models up to the trillion-parameter scale, including Kimi K2/2.5 (Team et al., 2025) and GLM-4.5/4.7 (Zeng et al., 2025).

---
[*]Equal contribution  [1]MBZUAI  [2]Nanjing University. Correspondence to: Xingyu Qu <Xingyu.Qu@mbzuai.ac.ae>, Samuel Horvath <Samuel.Horvath@mbzuai.ac.ae>.

*Proceedings of the $43^{rd}$ International Conference on Machine Learning*, Seoul, South Korea. PMLR 306, 2026. Copyright 2026 by the author(s).

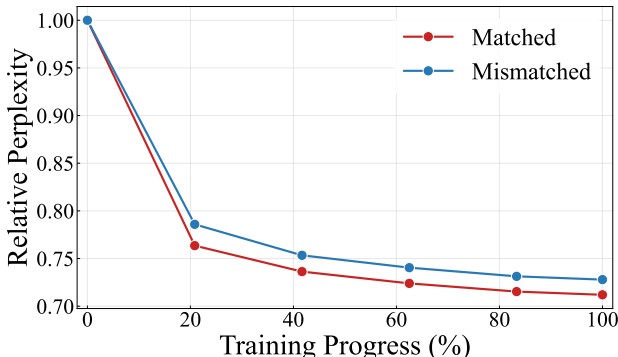

Figure 1. Relative perplexity (normalized by pretrained baseline) during full fine-tuning on NanoChat (Karpathy, 2025). Fine-tuning with a mismatched optimizer (e.g., using Muon on an Adam-pretrained model) consistently results in worse perplexity. See Section 3 for details.

Despite these successes, existing work on Muon has focused almost exclusively on pretraining, leaving fine-tuning, the dominant training paradigm, largely unexplored. Preliminary results from Liu et al. (2025) reveal an *optimizer mismatch* problem: applying Muon to fine-tune an Adam-pretrained model yields suboptimal results compared to Adam, and vice versa. We illustrate this in Figure 1. Since most open models are pretrained with Adam, this mismatch severely limits Muon's practical applicability. Understanding and addressing this mismatch is therefore critical.

This work presents the first in-depth analysis of the optimizer mismatch problem, combining empirical exploration with theoretical insights. We first reproduce the phenomenon through controlled experiments and relate it to the distinct implicit biases of Adam and Muon, which produce pretrained weights with different structural properties. We find that mismatch increases sensitivity to update strength during fine-tuning, suggesting that it degrades performance by disrupting pretrained knowledge. Based on this analysis, we hypothesize that constraining the extent of updates should mitigate the mismatch. We examine this hypothesis with Low-Rank Adaptation (LoRA) (Hu et al., 2022), which freezes the pretrained weights and restricts updates to a low-rank subspace, thereby limiting how much the fine-tuning optimizer can alter them. We verify this through extensive experiments on language and vision benchmarks, showing that LoRA combined with Muon (LoRA-Muon) matches

or outperforms its Adam counterpart (LoRA-Adam). We further validate our hypothesis through rank studies, catastrophic forgetting (McCloskey & Cohen, 1989; French, 1999) measurements, and investigation of LoRA variants.

To summarize, we make the following contributions:

- We reproduce and analyze the optimizer mismatch problem at accessible scales, relate it to the distinct implicit biases of Adam and Muon, and provide evidence that mismatch degrades performance by disrupting pretrained knowledge.

- We show that constraining updates via LoRA mitigates this mismatch, enabling LoRA-Muon to perform on-par with LoRA-Adam across language and vision tasks. Studies on LoRA rank, catastrophic forgetting, and compatibility with LoRA variants further support this finding.

## 2. Preliminaries

### 2.1. Muon Optimizer

Muon (Jordan et al., 2024) is an optimizer designed for matrix-shaped parameters in neural networks, and is typically paired with Adam for non-matrix parameters such as embeddings and biases. In practice, Muon's implementation varies slightly across frameworks, as detailed in Appendix B. We adopt the implementation of Liu et al. (2025) (except in Section 3), who first reported the optimizer mismatch problem. Their implementation serves as the basis for MuonClip, the optimizer used to pretrain the 32B/1T Kimi K2 model (Team et al., 2025).

Given a parameter matrix $W_t \in \mathbb{R}^{m \times n}$ and its gradient $G_t$, Muon updates the parameters as:

$$O_t = \text{NS}(M_t), \quad M_t = \beta M_{t-1} + G_t, \quad (1)$$

$$W_{t+1} = W_t - \eta O_t \quad (2)$$

where $\beta$ is the momentum coefficient, $\eta$ is the learning rate, and $\text{NS}(\cdot)$ denotes the Newton-Schulz iteration that approximates the nearest semi-orthogonal matrix. Specifically, for the singular value decomposition $M_t = U \Sigma V^\top$, we have $O_t \approx U V^\top = (M_t M_t^\top)^{-1/2} M_t$. This orthogonalization ensures that updates have nearly uniform singular values, effectively applying equal step sizes across all directions in the weight space. Later works, such as Polar Express (PE) (Amsel et al., 2026), replace the fixed Newton-Schulz coefficients with adaptive ones for a more accurate approximation.

In contrast, Adam (Kingma & Ba, 2015) is the dominant optimizer for both pretraining and fine-tuning large language models. It uses element-wise adaptive learning rates based on first- and second-moment estimates of the gradient:

$$M_t = \beta_1 M_{t-1} + (1 - \beta_1) G_t, \quad (3)$$

$$V_t = \beta_2 V_{t-1} + (1 - \beta_2) G_t^2, \quad (4)$$

$$W_{t+1} = W_t - \eta \cdot \hat{M}_t \oslash (\sqrt{\hat{V}_t} + \epsilon) \quad (5)$$

where $\hat{M}_t = M_t/(1 - \beta_1^t)$ and $\hat{V}_t = V_t/(1 - \beta_2^t)$ are the bias-corrected estimates, and $\oslash$ denotes element-wise division. The key distinction between these two optimizers lies in their preconditioning: Adam adapts step sizes independently for each parameter via element-wise rescaling $1/(\sqrt{\hat{V}_t} + \epsilon)$, whereas Muon adapts step sizes across singular directions of the gradient matrix via matrix-level preconditioning $(M M^\top)^{-1/2}$. As we show in Section 3, this difference leads to fundamentally different implicit biases, which in turn give rise to the optimizer mismatch problem.

### 2.2. Low-Rank Adaptation (LoRA)

Low-Rank Adaptation (LoRA) (Hu et al., 2022) is a parameter-efficient fine-tuning method that freezes the pretrained weights and introduces trainable low-rank decomposition matrices. For a pretrained weight matrix $W_0 \in \mathbb{R}^{m \times n}$, LoRA parameterizes the weight update as:

$$W = W_0 + \Delta W = W_0 + \tilde{\alpha} B A \quad (6)$$

where $B \in \mathbb{R}^{m \times r}$ and $A \in \mathbb{R}^{r \times n}$ are the trainable low-rank matrices, $r \ll \min(m, n)$ is the rank, and $\tilde{\alpha}$ is a scaling factor typically set to $\alpha/r$ (Hu et al., 2022) or $\alpha/\sqrt{r}$ (Kalajdzievski, 2023), where $\alpha$ is a hyperparameter. By default, $B$ is initialized to zero and $A$ is drawn from a random Gaussian, so that $\Delta W = 0$ at the start of training. During fine-tuning, only $A$ and $B$ are updated while $W_0$ remains frozen. This approach significantly reduces the number of trainable parameters and memory requirements.

However, LoRA often underperforms full fine-tuning due to the low-rank constraint. Various variants have been proposed to narrow this gap, including initialization techniques that bring LoRA updates closer to full fine-tuning (Zhang et al., 2025; Meng et al., 2024; Tastan et al., 2026). On the other hand, LoRA has been shown to "learn less but forget less," suggesting that the low-rank constraint helps preserve pretrained knowledge (Biderman et al., 2024).

## 3. Analyzing Optimizer Mismatch

Liu et al. (2025) reported that fine-tuning with a mismatched optimizer—using Adam on Muon-pretrained models or vice versa—leads to degraded performance compared to using the same optimizer for both stages. This *optimizer mismatch* problem significantly limits the practical applicability of Muon for fine-tuning, since most publicly available pretrained models were trained with Adam. To understand

*Table 1.* WikiText-2 validation perplexity normalized by the pretrained model baseline (averaged over 3 seeds). Best results per column are in **bold**. Blue rows indicate Muon fine-tuning. Red values show the gap from the matched optimizer, which is reduced with LoRA. Raw values are in Appendix C.

| Method | Muon Pretrain ↓ | Adam Pretrain ↓ |
|---|---|---|
| Full-Muon | **0.716** | 0.719 (+.009) |
| Full-Adam | 0.739 (+.023) | **0.710** |
| LoRA-Muon | 0.719 | 0.723 (+.002) |
| LoRA-Adam | 0.733 (+.014) | 0.721 |

this phenomenon, we conduct controlled experiments in a simplified setting.

**Experimental setup.** We pretrain two 561M-parameter NanoChat models (Karpathy, 2025) from scratch on ~11B tokens from FineWeb-Edu (Penedo et al., 2024) following the Chinchilla scaling law (Hoffmann et al., 2022): one with Muon and one with Adam (tuned to achieve similar CORE metrics (Li et al., 2024b)). We then fine-tune on WikiText-2 (Merity et al., 2017) using full fine-tuning and LoRA ($r = 8$, $\alpha = 16$), each with both optimizers (denoted Full-Muon, Full-Adam, LoRA-Muon, and LoRA-Adam), and report the best validation perplexity over a learning rate sweep (averaged over 3 seeds). Details on model architecture and experiments are in Appendix C.

**Reproducing the mismatch.** Table 1 confirms the optimizer mismatch phenomenon: for both pretrained models, using the matched optimizer (Full-Muon for Muon-pretrained, Full-Adam for Adam-pretrained) consistently outperforms the mismatched one. This symmetric pattern indicates a fundamental incompatibility between Muon and Adam when switching optimizers across pretraining and fine-tuning.

### 3.1. Why Does Mismatch Occur?

We hypothesize that the mismatch arises from the fundamentally different *implicit biases* of Adam and Muon. Specifically, Adam uses element-wise preconditioning, while Muon uses $(MM^\top)^{-1/2}$ for matrix-level preconditioning. This results in different implicit biases toward the max-norm $\|W\|_{\max} = \max_{i,j} |W_{ij}|$ and the spectral norm $\|W\|_2 = \sigma_{\max}(W)$, respectively. Bernstein & Newhouse (2024) interpret Adam and Muon (without momentum) as steepest descent under the above norms. On classification problems, Zhang et al. (2024); Fan et al. (2025) show that Adam converges to solutions with maximal max-norm margin, while Muon converges to solutions with maximal spectral-norm margin. Additionally, Chen et al. (2026) shows that Muon optimizes a spectral-norm constrained problem, and Kovalev (2025) characterizes it as a trust-region method in spectral norm.

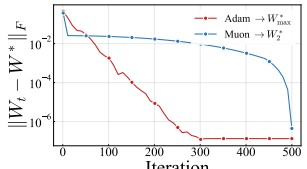 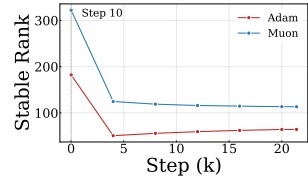

*Figure 2.* **Left:** Numerical verification of the implicit biases on a toy linear regression problem. Adam converges to the min-max-norm solution $W_{\max}^*$, while Muon converges to the min-spectral-norm solution $W_2^*$. **Right:** Average stable rank of Q, K, V projections during NanoChat pretraining. Muon-trained weights maintain notably higher stable rank, indicating a distinct spectral structure.

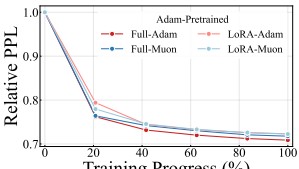 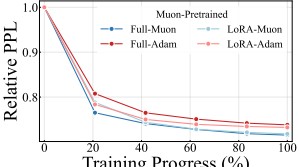

*Figure 3.* Fine-tuning perplexity (PPL) trajectories on Adam-pretrained (left) and Muon-pretrained (right) NanoChat models. LoRA mitigates the optimizer mismatch in both cases.

To further illustrate this, we analyze a simplified linear regression problem: minimizing $\mathcal{L}(W) = \frac{1}{2}\|Wx - y\|_2^2$ for $W \in \mathbb{R}^{m \times n}$, given $x \in \mathbb{R}^n$ and $y \in \mathbb{R}^m$, which allows closed-form tracking of the optimization dynamics. For simplicity, we consider Muon with exact orthogonalization and without momentum, and analyze the dynamics of SignGD as a simple yet insightful proxy of Adam (Balles & Hennig, 2018; Bernstein et al., 2018). In this setting, we show that the two optimizers converge to fundamentally different solutions (Theorems 3.1 and 3.2; proofs in Appendix D). Figure 2 (left) illustrates this numerically; see Appendix D for the corresponding loss curves.

**Theorem 3.1** (Implicit Bias of SignGD). *Consider SignGD from $W_0 = 0$ with step sizes $\eta_t \to 0$ and $\sum_t \eta_t = \infty$. The iterates converge to $W^* = y \,\mathrm{sign}(x)^\top / \|x\|_1$, which achieves the minimum max-norm among all solutions: $\|W^*\|_{\max} = \min_{W:Wx=y} \|W\|_{\max}$.*

**Theorem 3.2** (Implicit Bias of Muon). *Consider Muon from $W_0 = 0$ with step sizes $\eta_t \to 0$ and $\sum_t \eta_t = \infty$. The iterates converge to $W^* = yx^\top / \|x\|_2^2$, which achieves the minimum spectral norm among all solutions: $\|W^*\|_2 = \min_{W:Wx=y} \|W\|_2$.*

Beyond this simplified setting, these different implicit biases also lead to structurally different weights in practice. As shown in Figure 2 (right), Muon-trained weights exhibit notably higher stable rank during NanoChat pretraining; see Appendix F.1 for additional spectral analysis, including SVD entropy. Similar observations were reported by Liu et al. (2025) on larger-scale models.

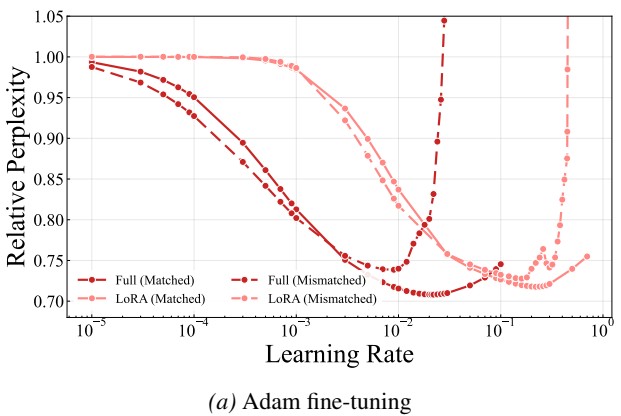

*(a)* Adam fine-tuning

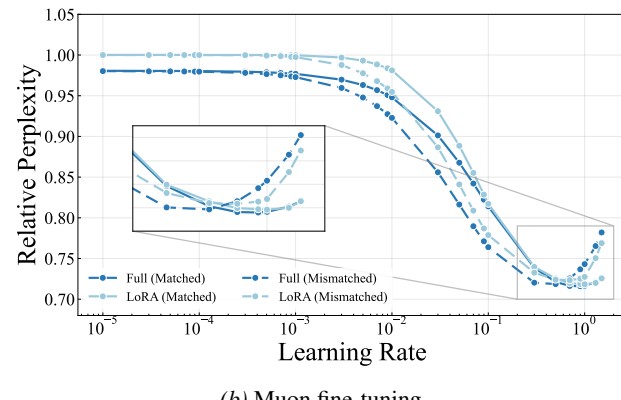

*(b)* Muon fine-tuning

*Figure 4.* Learning rate sweeps for fine-tuning with Adam (left) and Muon (right). Solid lines: matched pretraining optimizer. Dashed lines: mismatched pretraining optimizer. Dark colors: full fine-tuning. Light colors: LoRA. Under mismatch, the curve shifts upward and leftward (worse perplexity at a lower optimal learning rate). LoRA reduces the gap between matched and mismatched curves.

**Impact on fine-tuning.** Given these structural differences, fine-tuning with a mismatched optimizer can alter the pretrained weights in a direction incompatible with the pretraining structure, potentially disrupting the model's learned knowledge. Figure 4 provides evidence for this through learning rate sweeps: using a mismatched pretrained model shifts the perplexity curve *upward and leftward*—the optimal learning rate becomes smaller, and the best achievable perplexity is worse. This indicates that the model is more sensitive to update strength under mismatch, and that stronger updates cause more disruption to the pretrained knowledge. We further corroborate this by measuring catastrophic forgetting in Section 4.5.

### 3.2. LoRA Mitigates Optimizer Mismatch

Given that mismatch makes the model more sensitive to update strength, we hypothesize that *constraining the extent of updates during fine-tuning should mitigate the mismatch problem.*

We examine this idea with LoRA (Hu et al., 2022), which naturally achieves this through two mechanisms: (1) it preserves the pretrained weights $W_0$ exactly, optimizing only the low-rank adapters; (2) the low-rank constraint inherently limits the extent of updates. This aligns with recent findings that LoRA "learns less and forgets less" (Biderman et al., 2024). We formalize this intuition in the toy linear regression framework of Section 3.1: under LoRA-style constraints, the worst-case mismatch inflation is bounded by a factor that scales with rank $r$, vanishing at $r = 1$ and recovering full fine-tuning when $r = n$ (Appendix D.3).

Table 1 and Figure 3 confirm that LoRA reduces the mismatch gap: the perplexity gap shrinks by 39% for Muon-pretrained models and 78% for Adam-pretrained models. Notably, LoRA-Adam even outperforms Full-Adam on

Muon-pretrained models. On Adam-pretrained models, LoRA-Muon converges faster than LoRA-Adam early on, suggesting that Muon's fast early convergence transfers to fine-tuning under LoRA. Figure 4 provides further evidence: LoRA (light colors) narrows the gap between matched and mismatched curves, allowing larger learning rates under mismatch.

## 4. Experiments

Section 3 showed that optimizer mismatch disrupts pretrained knowledge, and that constraining updates via LoRA mitigates this effect. We now validate this hypothesis on standard benchmarks across natural language understanding (NLU), natural language generation (NLG), and image classification, examining whether the performance gap between Adam and Muon under full fine-tuning diminishes when LoRA is applied.

**Implementation.** Following the standard Muon implementation (Jordan et al., 2024; Liu et al., 2025), we use Nesterov momentum and shape-dependent learning rate scaling (see Appendix B). As Muon requires operating on full gradient matrices for Newton-Schulz orthogonalization, it is incompatible with standard distributed training frameworks such as Fully Sharded Data Parallel (FSDP) and DeepSpeed ZeRO (Rajbhandari et al., 2020) that shard tensors across devices. While recent work has proposed distributed Muon variants (Liu et al., 2025; Ahn et al., 2025; Li et al., 2025b), these are either not mathematically equivalent to the original Muon or not publicly available. To ensure a fair comparison, we use standard DDP (Distributed Data Parallel) training for all experiments with both Muon and Adam. The only exception is Full-Adam fine-tuning of Llama 2-7B (Touvron et al., 2023), where we use DeepSpeed ZeRO-2 due to memory constraints.

*Table 2.* Natural language understanding results on GLUE benchmark with T5-Base. We compare the performance gap between Adam and Muon under full fine-tuning versus LoRA. Results are accuracy (%) reported as mean ± std over 3 seeds. Best results are underlined; best LoRA results are **bolded**. Blue rows indicate Muon-based methods.

| Method | CoLA | MNLI | MRPC | QNLI | SST-2 | Average |
|---|---|---|---|---|---|---|
| Full-Adam | $82.90_{\pm0.40}$ | $\underline{86.61}_{\pm0.15}$ | $87.99_{\pm0.88}$ | $93.04_{\pm0.22}$ | $\underline{95.15}_{\pm0.29}$ | 89.14 |
| Full-Muon | $82.42_{\pm0.63}$ | $86.24_{\pm0.09}$ | $87.75_{\pm0.53}$ | $92.95_{\pm0.11}$ | $94.50_{\pm0.09}$ | 88.77 |
| Full-Muon-PE | $81.82_{\pm0.52}$ | $86.23_{\pm0.11}$ | $\underline{88.73}_{\pm1.22}$ | $92.96_{\pm0.10}$ | $94.88_{\pm0.14}$ | 88.92 |
| LoRA-Adam | $82.81_{\pm0.39}$ | $86.12_{\pm0.05}$ | $88.24_{\pm0.60}$ | $\mathbf{93.22}_{\pm0.05}$ | $94.27_{\pm0.16}$ | 88.93 |
| LoRA-Muon | $82.52_{\pm0.35}$ | $86.41_{\pm0.08}$ | $88.15_{\pm0.83}$ | $93.18_{\pm0.07}$ | $94.57_{\pm0.24}$ | 88.97 |
| LoRA-Muon-PE | $\mathbf{83.00}_{\pm0.23}$ | $\mathbf{86.43}_{\pm0.17}$ | $\mathbf{88.48}_{\pm0.72}$ | $93.14_{\pm0.05}$ | $\mathbf{94.95}_{\pm0.09}$ | $\underline{\mathbf{89.20}}$ |

*Table 3.* Natural language generation results with Llama 2-7B. We compare the performance gap between Adam and Muon under full fine-tuning versus LoRA on math (GSM8K accuracy), code (HumanEval Pass@1), and commonsense reasoning (average accuracy). Results are reported as mean ± std over 3 seeds. Best results are underlined; best LoRA results are **bolded**. Blue rows indicate Muon-based methods.

| Method | Math | Code | Commonsense | Average |
|---|---|---|---|---|
| Full-Adam | $\underline{61.66}_{\pm0.04}$ | $\underline{35.57}_{\pm1.37}$ | $67.52_{\pm0.07}$ | $\underline{54.92}$ |
| Full-Muon | $57.37_{\pm0.8}$ | $34.35_{\pm0.76}$ | $67.57_{\pm0.07}$ | 53.10 |
| Full-Muon-PE | $57.90_{\pm0.47}$ | $35.16_{\pm0.76}$ | $\underline{67.62}_{\pm0.12}$ | 53.56 |
| LoRA-Adam | $\mathbf{59.64}_{\pm0.66}$ | $27.85_{\pm1.44}$ | $67.37_{\pm0.11}$ | 51.62 |
| LoRA-Muon | $59.57_{\pm0.4}$ | $\mathbf{29.47}_{\pm0.76}$ | $\mathbf{67.40}_{\pm0.11}$ | **52.15** |
| LoRA-Muon-PE | $59.24_{\pm0.66}$ | $28.86_{\pm1.75}$ | $67.36_{\pm0.12}$ | 51.82 |

### 4.1. Natural Language Understanding

**Setup.** Following prior work (Wang et al., 2024; Zhang et al., 2025), we evaluate on T5-Base (Raffel et al., 2020) fine-tuned on GLUE (Wang et al., 2019) tasks (CoLA, MNLI, MRPC, QNLI, SST-2). T5 was pretrained with Adafactor (Shazeer & Stern, 2018), a memory-efficient variant of Adam. We apply LoRA with rank $r = 8$ and $\alpha = 16$ to all linear layers except embeddings and the language model head. We train for 5 epochs on MRPC and CoLA, and 3 epochs on SST-2, QNLI, and MNLI. We perform a learning rate sweep for each method on each dataset and report results averaged over 3 seeds. Full experimental details are provided in Appendix E.1.

**Results.** Table 2 presents the results. As all methods are well-tuned and trained to near-convergence, absolute differences are modest. Nevertheless, for full fine-tuning, Muon still underperforms Adam, consistent with the optimizer mismatch phenomenon. Under LoRA, however, the gap disappears: LoRA-Muon slightly outperforms LoRA-Adam, and LoRA-Muon-PE (Muon with PE coefficients) achieves the highest average accuracy among all methods, surpassing even Full-Adam. For full fine-tuning, PE also improves Muon, though a gap with Adam remains. These results support our hypothesis: LoRA effectively mitigates optimizer mismatch, transforming Muon from underperforming Adam under full fine-tuning to matching or outperforming it under LoRA.

### 4.2. Natural Language Generation

**Setup.** Following prior work (Wang et al., 2024; Zhang et al., 2025), we instruction-tune Llama 2-7B, an Adam-pretrained model, on three tasks: math, code, and commonsense reasoning. For math, we use a 100k subset of MetaMathQA (Yu et al., 2024) bootstrapped from GSM8K (Cobbe et al., 2021), and evaluate accuracy on the GSM8K test set. For code, we use a 100k subset of Code-Feedback (Zheng et al., 2024), and report Pass@1 on HumanEval (Chen et al., 2021). For commonsense reasoning, we instruction-tune on a 52k subset of WizardLM (Xu et al., 2024), and evaluate on commonsense reasoning benchmarks (ARC (Clark et al., 2018), HellaSwag (Zellers et al., 2019), PIQA (Bisk et al., 2020), WinoGrande (Sakaguchi et al., 2020), BoolQ (Clark et al., 2019), OpenBookQA (Mihaylov et al., 2018)) using lm-evaluation-harness (Gao et al., 2024). All models are trained for 1 epoch. For LoRA methods, we use rank $r = 8$ and $\alpha = 16$. For HumanEval and GSM8K evaluation, we use greedy decoding. We perform a learning rate sweep for each method and task and report results at the final checkpoint, averaged over 3 seeds. Full experimental details are provided in Appendix E.2.

**Results.** Table 3 presents the results. For full fine-tuning, Muon underperforms Adam, particularly on math, with a

*Table 4.* Image classification results with CLIP ViT-B/32. We compare the performance gap between Adam and Muon under full fine-tuning versus LoRA. Results are accuracy (%) reported as mean ± std over 3 seeds. Best results are underlined; best LoRA results are **bolded**. Blue rows indicate Muon-based methods.

| Method | StanfordCars | DTD | GTSRB | RESISC45 | SUN397 | SVHN | Average |
|---|---|---|---|---|---|---|---|
| Full-Adam | $78.02_{\pm0.33}$ | $\underline{75.49}_{\pm0.57}$ | $98.85_{\pm0.09}$ | $\underline{95.10}_{\pm0.14}$ | $\underline{74.83}_{\pm0.13}$ | $97.04_{\pm0.12}$ | $\underline{86.55}_{\pm0.23}$ |
| Full-Muon | $\underline{79.41}_{\pm0.74}$ | $72.39_{\pm0.54}$ | $98.46_{\pm0.15}$ | $94.58_{\pm0.13}$ | $74.21_{\pm0.25}$ | $97.25_{\pm0.06}$ | $86.05_{\pm0.31}$ |
| Full-Muon-PE | $78.31_{\pm0.33}$ | $72.93_{\pm0.11}$ | $98.55_{\pm0.12}$ | $94.68_{\pm0.31}$ | $73.83_{\pm0.19}$ | $\underline{97.34}_{\pm0.08}$ | $85.94_{\pm0.19}$ |
| LoRA-Adam | $72.50_{\pm0.16}$ | $\mathbf{73.88}_{\pm0.61}$ | $98.48_{\pm0.10}$ | $94.41_{\pm0.14}$ | $\mathbf{68.86}_{\pm0.08}$ | $96.87_{\pm0.09}$ | $84.17_{\pm0.20}$ |
| LoRA-Muon | $74.95_{\pm0.27}$ | $73.24_{\pm0.59}$ | $98.65_{\pm0.12}$ | $94.97_{\pm0.30}$ | $68.10_{\pm0.28}$ | $\mathbf{96.96}_{\pm0.08}$ | $84.48_{\pm0.27}$ |
| LoRA-Muon-PE | $\mathbf{75.45}_{\pm0.61}$ | $73.64_{\pm1.10}$ | $\mathbf{98.91}_{\pm0.22}$ | $\mathbf{94.98}_{\pm0.09}$ | $68.40_{\pm0.08}$ | $96.91_{\pm0.03}$ | $\mathbf{84.71}_{\pm0.35}$ |

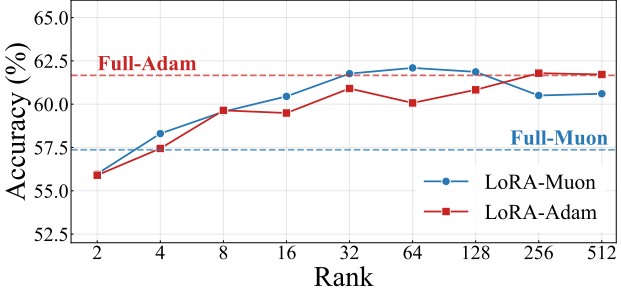

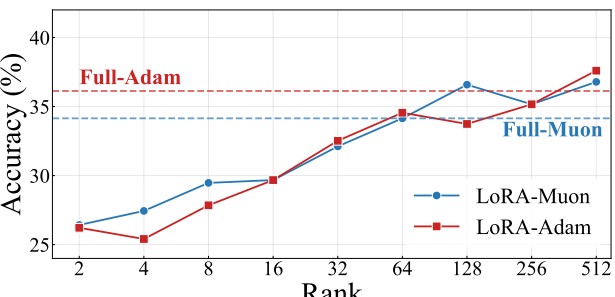

*(a)* Fine-tuned on MetaMath, evaluated on GSM8K.      *(b)* Fine-tuned on Code-Feedback, evaluated on HumanEval.

*Figure 5.* Effect of LoRA rank on downstream performance when fine-tuning Llama 2-7B. Dashed lines indicate full fine-tuning performance. When mismatch is pronounced (a), LoRA-Muon outperforms LoRA-Adam at low to moderate ranks but degrades at high ranks as updates increasingly resemble full fine-tuning. When the mismatch is mild (b), LoRA-Muon performs comparably across all ranks.

smaller gap on code and negligible differences on commonsense reasoning. Under LoRA, the gap largely disappears: LoRA-Muon matches LoRA-Adam on math and outperforms it on code and commonsense reasoning. These results are consistent with our NLU findings, confirming that LoRA enables Muon to match or surpass Adam across different tasks and models. We observe the same pattern on Llama 2-13B (Appendix E.2). Interestingly, PE consistently improves full fine-tuning but slightly degrades LoRA performance, suggesting that the more accurate orthogonalization in PE does not necessarily benefit the LoRA setting.

### 4.3. Image Classification

**Setup.** Following prior work (Li et al., 2025a; Wang et al., 2025; He et al., 2025), we fine-tune CLIP ViT-B/32 (Radford et al., 2021), an Adam-pretrained model, on six image classification tasks: StanfordCars (Krause et al., 2013), DTD (Cimpoi et al., 2014), GTSRB (Stallkamp et al., 2011), RESISC45 (Cheng et al., 2017), SUN397 (Xiao et al., 2010), and SVHN (Netzer et al., 2011). We freeze the CLIP text tower and adapt the vision tower via full fine-tuning and LoRA with $r = 8$ and $\alpha = 16$. We perform a learning rate sweep for each method, and report results averaged over 3 seeds. Full experimental details are provided in Ap-

pendix E.3.

**Results.** Table 4 reports the results. Unlike the language tasks, the full fine-tuning gap between Adam and Muon is small in this vision setting. Under LoRA, Muon and Muon-PE both outperform Adam on average, suggesting that LoRA's mismatch mitigation effect extends to vision tasks.

**Statistical significance across tasks.** We assess the statistical significance of LoRA's mismatch mitigation by computing the reduction in the Adam–Muon performance gap when switching from full fine-tuning to LoRA for each task, and aggregating across all tasks in Tables 2–4 using random-effects meta-analysis. The pooled gap reduction is 0.72% (95% CI: [0.41, 1.04], $p < 0.001$) for Muon and 0.83% (95% CI: [0.45, 1.20], $p < 0.001$) for Muon-PE, confirming that LoRA significantly mitigates the optimizer mismatch across tasks.

### 4.4. Effect of LoRA Rank

Our analysis in Section 3 suggests that LoRA mitigates optimizer mismatch by limiting updates to the pretrained weights. A natural prediction is that this benefit may diminish at higher ranks, as LoRA increasingly resembles full

*Table 5.* Commonsense reasoning performance after fine-tuning Llama 2-7B on MetaMath. Lower performance indicates more severe catastrophic forgetting. Results are averaged over 3 seeds. Pretrained baseline is in **bold**; best full fine-tuning results are boxed; best LoRA results are underlined. Blue rows indicate Muon-based methods.

| Method | ARC-c | ARC-e | HellaSwag | OBQA | PIQA | Average |
|---|---|---|---|---|---|---|
| Pretrained | **45.0** | **73.8** | **76.2** | **44.0** | **78.7** | **63.5** |
| Full-Adam | 36.3 | 56.7 | 72.3 | 42.2 | 76.6 | 56.8 |
| Full-Muon | 35.0 | 56.2 | 69.5 | 40.2 | 75.9 | 55.4 |
| Full-Muon-PE | 33.7 | 54.3 | 68.2 | 39.1 | 74.9 | 54.1 |
| LoRA-Adam | 37.3 | 56.0 | 74.2 | 43.9 | 77.0 | 57.7 |
| LoRA-Muon | 36.7 | 54.8 | 73.2 | 43.3 | 76.7 | 56.9 |
| LoRA-Muon-PE | 37.1 | 54.3 | 72.5 | 41.7 | 76.3 | 56.4 |

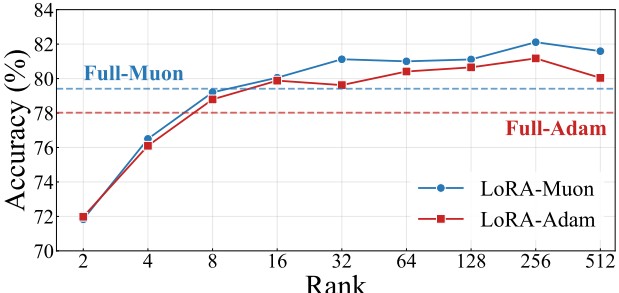
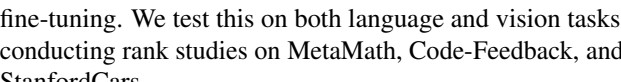

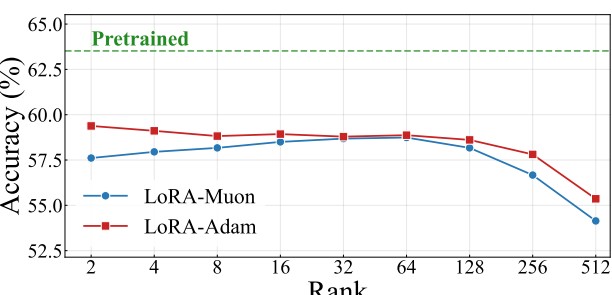

*Figure 6.* Effect of LoRA rank on accuracy when fine-tuning CLIP ViT-B/32 on StanfordCars. LoRA-Muon outperforms LoRA-Adam across nearly all ranks ($r \geq 4$). Dashed lines indicate full fine-tuning performance.

*Figure 7.* Effect of LoRA rank on catastrophic forgetting when fine-tuning Llama 2-7B on MetaMath, measured by commonsense reasoning accuracy. Lower accuracy indicates more severe forgetting.

fine-tuning. We test this on both language and vision tasks, conducting rank studies on MetaMath, Code-Feedback, and StanfordCars.

**Setup.** We vary the LoRA rank across $r \in \{2, 4, 8, 16, 32, 64, 128, 256, 512\}$ with $\alpha = 2r$. For each rank, we perform a learning rate sweep and report the best result averaged over 3 seeds. Other settings follow Sections 4.2 and 4.3.

**Results.** Figures 5a, 5b, and 6 present the results. On Meta-Math (Figure 5a), LoRA-Muon matches or outperforms LoRA-Adam at low to moderate ranks. At higher ranks, however, LoRA-Muon begins to degrade while LoRA-Adam continues to improve—consistent with our hypothesis, as higher-rank updates increasingly resemble full fine-tuning where mismatch is most severe. On Code-Feedback (Figure 5b), where the mismatch is milder, LoRA-Muon performs comparably to LoRA-Adam across all ranks. On StanfordCars (Figure 6), where the mismatch is also mild, LoRA-Muon outperforms LoRA-Adam across nearly all ranks, with the advantage widening at higher ranks.

These results support our hypothesis: constraining the extent of updates mitigates optimizer mismatch. When mismatch is pronounced, this constraint is beneficial at low ranks but

becomes insufficient at high ranks as the updates increasingly resemble full fine-tuning. When the mismatch is mild, LoRA-Muon performs well across all ranks, as Muon can leverage its faster convergence (Liu et al., 2025).

### 4.5. Measuring Catastrophic Forgetting

In Section 3, we hypothesized that optimizer mismatch degrades performance by disrupting pretrained knowledge. Catastrophic forgetting (McCloskey & Cohen, 1989; French, 1999) provides a direct way to test this hypothesis. Following Kotha et al. (2024), we measure forgetting by fine-tuning on MetaMath and evaluating on commonsense reasoning benchmarks. We use the Llama 2-7B models fine-tuned in Section 4.2 and exclude tasks where forgetting is negligible under full fine-tuning, as they are not informative for our analysis (see Appendix E.5 for details).

**Results.** Table 5 shows commonsense benchmark performance after math fine-tuning. Full fine-tuning causes substantial forgetting for both optimizers, but Full-Muon forgets more than Full-Adam despite achieving worse fine-tuning performance (Table 3). This suggests that the mismatch does not simply lead to weaker learning but actively disrupts pretrained knowledge. LoRA methods preserve pretrained

*Table 6.* Comparison of LoRA variants on GLUE benchmark with T5-Base. All methods use the same training setup and learning rate sweep. Results are test accuracy (%) averaged over 3 seeds. Best results are **bolded**. Blue rows indicate Muon-based methods.

| Method | CoLA | MNLI | MRPC | QNLI | SST-2 | Average |
|---|---|---|---|---|---|---|
| LoRA-Adam | 82.81 | 86.12 | 88.24 | 93.22 | 94.27 | 88.93 |
| LoRA-Muon-PE | 83.00 | 86.43 | 88.48 | 93.14 | **94.95** | **89.20** |
| rsLoRA-Adam (Kalajdzievski, 2023) | 83.09 | 86.12 | 88.48 | 93.11 | 94.76 | 89.11 |
| LoRA-One-Adam (Zhang et al., 2025) | **83.32** | 86.16 | 88.48 | 93.25 | 94.61 | 89.16 |
| PiSSA-Adam (Meng et al., 2024) | 82.81 | 86.26 | 88.32 | 93.01 | 94.38 | 88.95 |
| rsLoRA-Muon-PE | 83.22 | 86.20 | 88.48 | 93.16 | 94.53 | 89.12 |
| LoRA-One-Muon-PE | 83.19 | **86.49** | 88.32 | 93.15 | 94.30 | 89.09 |
| PiSSA-Muon-PE | 82.84 | 86.19 | **89.79** | 92.79 | 94.00 | 89.12 |
| AdaLoRA-Adam (Zhang et al., 2023) | 82.49 | 85.68 | 86.48 | **93.29** | 94.38 | 88.46 |
| LoRA-Pro-Adam (Wang et al., 2025) | 82.81 | 86.23 | 88.56 | 93.15 | 94.80 | 89.11 |
| LoRA-RITE-Adam (Yen et al., 2025) | 83.09 | 86.47 | 87.99 | 93.17 | 94.27 | 89.00 |
| DoRA-Adam (yang Liu et al., 2024) | 82.74 | 86.28 | 88.24 | 93.21 | 94.57 | 89.01 |

knowledge better, with LoRA-Muon showing less forgetting than Full-Muon, supporting our hypothesis that constraining updates helps mitigate optimizer mismatch.

**Effect of Rank on Forgetting.** To examine how rank affects forgetting, we vary the LoRA rank while fixing the learning rate for fair comparison (Figure 7). For LoRA-Adam, forgetting increases steadily with rank, consistent with the observation that higher-rank LoRA learns more but also forgets more (Biderman et al., 2024). LoRA-Muon, however, initially shows *decreasing* forgetting as rank grows, narrowing the gap with LoRA-Adam until it nearly vanishes at moderate ranks ($r = 32$–$64$). This suggests that at low ranks, limited capacity leads to greater disruption of pretrained knowledge, whereas higher ranks provide more room to learn without as much forgetting. Beyond moderate ranks, both methods forget rapidly, but LoRA-Muon forgets faster as the effect of optimizer mismatch becomes more pronounced, consistent with Section 4.4.

**Weight Distance from Pretrained Model.** The forgetting analysis above measures disruption through downstream task performance. We complement this with a weight-space perspective: measuring the L2 and cosine distance between fine-tuned and pretrained weights for the models in Table 3 (full results in Appendix Table 17). Under full fine-tuning, Muon's cosine distance is 5.6–7.4× larger than Adam's on Math/Code, confirming a more aggressive departure from the pretrained structure. Under LoRA, this reverses: Muon's cosine distance is only 0.2–0.8× of Adam's. Notably, on Commonsense—where mismatch is mild, and Full-Muon already matches Full-Adam (Table 3)—Muon's distance is already smaller under full fine-tuning (0.65–0.75×), and LoRA further reduces it to 0.15–0.18×. This correlation between mismatch severity and weight displacement suggests that LoRA specifically preserves the pretrained structure

where mismatch would otherwise disrupt it. We also provide a spectral analysis of the LoRA matrices themselves during fine-tuning (Appendix F.2), showing that Muon's implicit bias toward uniform singular values manifests in the adapter weights.

### 4.6. Investigating LoRA Variants

Having shown that LoRA enables Muon to be effective for fine-tuning, we examine whether existing LoRA variants, typically evaluated with Adam, are also compatible with Muon. We evaluate on our NLU benchmark (Section 4.1) using identical training settings and the same learning rate sweep for each method. As PE consistently improves Muon-based methods on this benchmark, we report results with PE enabled for all Muon variants.

We first consider optimizer-agnostic variants that can be directly applied to LoRA-Muon: rsLoRA (Kalajdzievski, 2023), which modifies the LoRA scaling $\tilde{\alpha}$ from $\alpha/r$ to $\alpha/\sqrt{r}$ to stabilize training across different ranks; LoRA-One (Zhang et al., 2025), which initializes the LoRA matrices using a one-step gradient approximation to accelerate early convergence; and PiSSA (Meng et al., 2024), which initializes with the principal singular components of the pretrained weights to better preserve pretrained capabilities. We also compare against algorithm-modifying variants—AdaLoRA (Zhang et al., 2023), LoRA-Pro (Wang et al., 2025), LoRA-RITE (Yen et al., 2025), and DoRA (yang Liu et al., 2024)—which modify the training algorithm and are not directly compatible with Muon.

As shown in Table 6, the optimizer-agnostic variants improve LoRA-Adam, but applying them to LoRA-Muon-PE does not yield further improvement. This is consistent with our analysis in previous sections: rsLoRA increases the ef-

fective scaling $\tilde{\alpha}$, amplifying update magnitude and thus the mismatch effect (Section 3), while LoRA-One and PiSSA are designed to make LoRA updates closer to full fine-tuning, where mismatch is more severe (Section 4.4). These results suggest that existing LoRA variants may require adaptation to work effectively with Muon. Overall, LoRA-Muon-PE achieves the best average performance among all methods, outperforming even the algorithm-modifying variants that require additional memory and computation (e.g., LoRA-Pro, LoRA-RITE). This highlights Muon's potential as a competitive optimizer for LoRA fine-tuning.

### 4.7. Computational Efficiency

Tables 12 and 13 in the Appendix report wall-clock training time for Llama 2-7B and CLIP ViT-B/32, respectively. Under LoRA, where both optimizers use the same DDP framework, LoRA-Muon is only 1.1–1.2× slower than LoRA-Adam on Llama 2-7B and 1.0–1.1× on CLIP, indicating modest per-step overhead from the Newton-Schulz iteration. For full fine-tuning on Llama, the apparent gap is larger (2.3–2.9×), but this comparison is confounded by different distributed strategies: as noted at the start of Section 4, Full-Adam cannot fit in memory with DDP and requires DeepSpeed ZeRO-2, whereas Muon's lower memory footprint enables standard DDP. On CLIP, where both use single-GPU training and the comparison is direct, Full-Muon is only 1.0–1.2× slower. Indeed, Muon stores only one momentum buffer versus Adam's two (momentum + second moment), saving 50% of optimizer states ($\sim$14GB for Llama 2-7B in FP32)—this memory efficiency is itself a practical advantage. We note that Liu et al. (2025) find Muon $\sim$2× more efficient than Adam under compute-optimal pretraining, and as Muon's ecosystem matures, we expect the practical overhead to diminish further.

## 5. Discussion

We investigated the optimizer mismatch problem when switching between Adam and Muon across pretraining and fine-tuning, linking it to the distinct implicit biases of the two optimizers and showing that it degrades performance by disrupting pretrained knowledge. This insight led us to show that LoRA mitigates the issue, enabling LoRA-Muon to match or outperform LoRA-Adam across language and vision tasks. Studies on LoRA rank, catastrophic forgetting, and LoRA variants further supported our hypothesis. These findings shed light on how to leverage Muon's efficiency for fine-tuning without suffering from optimizer mismatch.

**Practical Recommendations.** For practitioners fine-tuning Adam-pretrained models with Muon, our results suggest: (1) under LoRA, Muon can serve as a drop-in replacement for Adam without performance loss, while saving 50% optimizer-state memory; (2) tune the learning rate sepa-

rately for Muon, as the optimal learning rate often differs from Adam's (Figure 4); (3) prefer moderate LoRA ranks that balance expressiveness against mismatch severity (Section 4.4); (4) do not assume that LoRA variants optimized for Adam transfer directly to Muon (Section 4.6).

**Limitations and Future Work.** While we empirically show that mismatched implicit biases disrupt pretrained knowledge, a theoretical characterization remains open; it may also be possible to reduce this structural gap before fine-tuning through specialized initialization or warmup. Due to resource constraints, our Muon-pretrained experiments are limited to NanoChat (561M); for Adam-pretrained models, our main experiments use 7B with preliminary results on 13B (Appendix E.2). Our experiments show that mismatch severity varies across tasks; understanding what factors determine this remains an open question. Beyond LoRA, our hypothesis suggests that other techniques for constraining updates—such as regularization or methods to mitigate catastrophic forgetting—may also help. Additionally, existing LoRA variants do not benefit LoRA-Muon, suggesting the need for Muon-specific adaptations.

## Impact Statement

This paper presents work aimed at advancing the field of Machine Learning. There are many potential societal consequences of our work, none of which we feel must be specifically highlighted here.

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

# Appendix

## A. Related Work

**Muon**    Muon was originally proposed by Jordan et al. (2024) as an optimizer for hidden layers in neural networks. Built upon SGD with momentum, Muon adds a per-layer orthogonalization step that projects the momentum matrix onto the set of semi-orthogonal matrices via Newton-Schulz iteration. This can be interpreted as steepest descent under the spectral norm (Bernstein & Newhouse, 2024), equalizing the contribution of all update directions. Muon can also be viewed as an "accumulation-free" variant of Shampoo (Gupta et al., 2018), where the preconditioner is computed from the current gradient alone rather than accumulated over training history.

Several works have validated Muon's scalability and efficiency. Shah et al. (2025) demonstrated that Muon extends the compute-time Pareto frontier and maintains data efficiency at batch sizes far exceeding the critical batch size, while requiring less memory than Adam due to storing only the first moment. Liu et al. (2025) showed that Muon achieves approximately $2\times$ compute efficiency over Adam in scaling law experiments, and introduced techniques, including weight decay and per-parameter update scaling that enable stable training at larger scales. Their Moonlight model (3B/16B MoE) was trained using these improvements. At an industrial scale, Muon and its variants have been adopted for training state-of-the-art large language models, including Kimi K2/2.5 (Team et al., 2025) and GLM-4.5/4.7 (Zeng et al., 2025), demonstrating its practical viability for frontier model development.

Recent work has proposed several improvements to the Muon algorithm. NorMuon (Li et al., 2025b) addresses the issue that orthogonalized updates exhibit high variance in per-neuron norms by adding neuron-wise normalization with adaptive learning rates, achieving further efficiency gains over vanilla Muon. Dion (Ahn et al., 2025) replaces Newton-Schulz iteration with amortized power iteration to better integrate with distributed training and weight sharding, reducing both computation and communication costs. On the algorithmic side, Amsel et al. (2026) and Grishina et al. (2025) proposed adaptive polynomial methods that accelerate the orthogonalization step, achieving faster convergence than the standard Newton-Schulz coefficients.

Theoretical understanding of Muon has also progressed. Kovalev (2025) provided the first convergence analysis by interpreting Muon as a trust-region method under the spectral norm, proving $O(1/\epsilon^4)$ iteration complexity for non-convex objectives. Su (2025) proposed an isotropic curvature model, suggesting that while Muon's direction of homogenizing singular values is correct, full orthogonalization may not be strictly optimal—spectrum homogenization suffices.

Despite these advances, existing work on Muon has focused almost exclusively on pretraining, leaving fine-tuning largely unexplored. Liu et al. (2025) first identified the *optimizer mismatch* problem: fine-tuning an Adam-pretrained model with Muon (or vice versa) leads to degraded performance, significantly limiting Muon's practical applicability given that most

open models are pretrained with Adam. This problem remains poorly understood and unresolved. This work presents the first systematic analysis of optimizer mismatch, combining theoretical insights with empirical investigation, and shows that constraining updates during fine-tuning, e.g., via LoRA, can mitigate this issue.

**Low-Rank Adaptation.** Low-Rank Adaptation (LoRA) (Hu et al., 2022) has become the dominant parameter-efficient fine-tuning method for large language models, enabling adaptation with significantly reduced memory and storage costs. However, LoRA often underperforms full fine-tuning due to the low-rank constraint on updates (Ivison et al., 2023; Biderman et al., 2024; Shuttleworth et al., 2025). Numerous variants have been proposed to narrow this gap, including initialization-based methods such as PiSSA (Meng et al., 2024), LoRA-GA (Wang et al., 2024), GoRA (He et al., 2025), and LoRA-One (Zhang et al., 2025), and algorithm-modifying variants like AdaLoRA (Zhang et al., 2023), DoRA (yang Liu et al., 2024), LoRA-Pro (Wang et al., 2025), and LoRA-RITE (Yen et al., 2025). Many of these methods aim to make LoRA updates closer to full fine-tuning trajectories. On the other hand, full fine-tuning can cause catastrophic forgetting (McCloskey & Cohen, 1989; French, 1999), where adapting to new tasks degrades the general capabilities acquired during pretraining (Vu et al., 2022; Kleiman et al., 2023; Kalajdzievski, 2024; Huang et al., 2024). Recent work showed that LoRA "learns less but forgets less" (Biderman et al., 2024), as the low-rank constraint limits the magnitude of weight changes and helps preserve pretrained knowledge. This property of LoRA — limiting weight changes while preserving pretrained knowledge — is central to our finding that LoRA can mitigate the optimizer mismatch problem.

## B. Muon Implementation

Several Muon implementations exist in the community, differing in two aspects: the scaling factor applied to the orthogonalized update and the momentum update rule. This section provides a brief introduction and explains our choice.

For a weight matrix $W \in \mathbb{R}^{m \times n}$, the two main implementation apply the following update rules:

- **Original** (Jordan et al., 2024): $W \leftarrow W - \eta \sqrt{\max(1, m/n)} \cdot \text{NS}(M_t)$

- **Moonlight** (Liu et al., 2025): $W \leftarrow W - 0.2\eta \sqrt{\max(m, n)} \cdot \text{NS}(M_t)$

where $\text{NS}(\cdot)$ denotes Newton-Schulz orthogonalization and $M$ is the momentum buffer, and $M_t$ is the updated momentum at step $t$.

The scaling factors differ in their dependence on matrix dimensions: the original Muon uses $\sqrt{\max(1, m/n)}$, which is sensitive to the ordering of $m$ and $n$, while the Moonlight variant uses $\sqrt{\max(m, n)}$, which is symmetric. For momentum, the original Muon uses an exponential moving average (EMA) update $M_t = \beta M_{t-1} + (1-\beta)G_t$, similar to Adam's first moment, while the Moonlight variant uses classical momentum $M_t = \beta M_{t-1} + G_t$. We adopt the Moonlight variant in our experiments. This choice is motivated by two factors: the optimizer mismatch problem was first identified using this implementation (Liu et al., 2025), and it has been successfully used to train large-scale models such as Kimi K2/2.5 (Team et al., 2025). Following standard practice (Jordan et al., 2024; Liu et al., 2025), we apply Muon only to two-dimensional weight matrices, while other parameters (e.g., biases, embeddings) are optimized with Adam.

Following standard Muon practice (Jordan et al., 2024; Liu et al., 2025), we also employ Nesterov-style momentum (Nesterov, 1983) in all our experiments.

For the orthogonalization step, the standard Newton-Schulz iteration uses fixed quintic polynomial coefficients $(a, b, c) = (3.4445, -4.7750, 2.0315)$ per step (Jordan et al., 2024). Polar Express (PE) (Amsel et al., 2026) replaces these with adaptive coefficients: for each iteration $i$, optimal coefficients $(a_i, b_i, c_i)$ are precomputed by solving for equioscillating quintic polynomials that minimize the approximation error on the interval $[\ell, 1]$, where $\ell$ is a lower bound on the smallest singular value ratio. This adaptive scheme accelerates convergence to the orthogonal matrix. We evaluate both standard Newton-Schulz (Muon) and the PE variant (Muon-PE) in our experiments.

## C. NanoChat Experiment

We use the NanoChat framework (Karpathy, 2025) for our controlled experiments in Section 3, including its Muon optimizer implementation, which follows the original Muon variant (Appendix B) without Polar Express. NanoChat implements a GPT-style (Vaswani et al., 2017; Radford et al., 2019) decoder-only Transformer with several modern architectural choices:

- **Architecture**: RoPE positional embeddings (Su et al., 2024), QK-norm for training stability (Henry et al., 2020), and ReLU$^2$ activation (So et al., 2021) instead of GELU (Hendrycks & Gimpel, 2023).

- **Model size**: Depth 20, hidden dimension $d = 1536$, 12 attention heads, resulting in approximately 561M parameters (the "\$100 model" configuration).

- **Pretraining data**: ~11B tokens from the FineWeb-Edu dataset (Penedo et al., 2024), following the Chinchilla-optimal 20:1 data-to-parameter ratio.

**Pretraining setup.** Both models are pretrained for 21,400 iterations with a total batch size of 524,288 tokens and a sequence length of 2048. Following NanoChat's design, we use different learning rates for different parameter groups. The "base" learning rates are scaled by $(d/768)^{-0.5}$ for embedding and unembedding parameters to account for model dimension.

- **Muon**: Matrix learning rate 0.02 (no scaling), embedding learning rate $0.2 \times (1536/768)^{-0.5} \approx 0.14$, unembedding learning rate $0.004 \times (1536/768)^{-0.5} \approx 0.003$.

- **Adam**: Matrix learning rate 1e-3 (with the same scaling applied to embedding/unembedding), tuned to achieve a comparable CORE (Li et al., 2024b) metric to Muon.

Table 7 shows the CORE metric for our pretrained models compared to GPT-2 Large (Radford et al., 2019). Figure 8 shows the training loss and validation BPB (bits per byte) curves during pretraining.

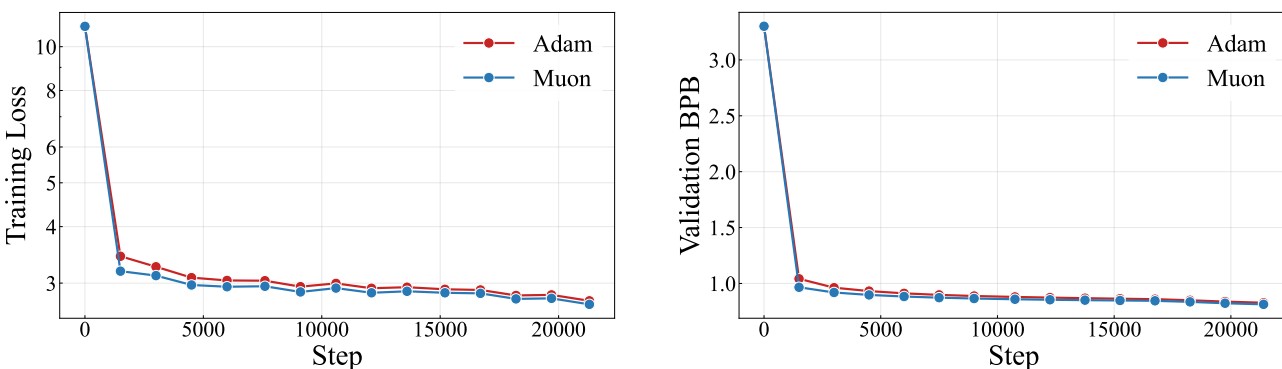

*Figure 8.* NanoChat pretraining curves. **Left:** Training loss. **Right:** Validation BPB (bits per byte). Both optimizers achieve similar final performance, with Muon converging slightly faster.

*Table 7.* CORE metric for pretrained models. GPT-2 Large score is from the NanoChat repository[1].

| Model | CORE ↑ |
|---|---|
| GPT-2 Large (774M) | 0.21 |
| NanoChat-Muon (561M) | 0.21 |
| NanoChat-Adam (561M) | 0.19 |

**Fine-tuning setup.** For fine-tuning on WikiText-2, we use:

- **Sequence length**: 1024

- **LoRA configuration**: Rank $r = 8$, $\alpha = 16$, dropout 0, applied to all attention projections (Q, K, V, output) and MLP projections (up, down).

- **Training**: 1 epoch, 3 random seeds per configuration.

---

[1] https://github.com/karpathy/nanochat/discussions/1

We sweep learning rates from 1e-5 to 9e-1 (with the same $(d/768)^{-0.5}$ scaling as pretraining) and report the best validation perplexity. Table 8 shows the selected learning rates for each configuration.

*Table 8.* Selected learning rates and validation perplexity for WikiText-2 fine-tuning.

| Pretrain | Fine-tune Method | Best LR | PPL ↓ |
|---|---|---|---|
| Muon | Full-Muon | 0.9 | **14.38** |
|  | Full-Adam | 0.009 | 14.84 |
|  | LoRA-Muon | 0.9 | 14.44 |
|  | LoRA-Adam | 0.1 | 14.72 |
| Adam | Full-Adam | 0.03 | **14.95** |
|  | Full-Muon | 0.5 | 15.13 |
|  | LoRA-Adam | 0.3 | 15.18 |
|  | LoRA-Muon | 0.7 | 15.23 |

## D. Theoretical Analysis: Implicit Bias of Optimizers

In this section, we analyze the implicit biases of Muon and SignGD on a simplified underdetermined linear regression problem. We use SignGD as a proxy for Adam (Balles & Hennig, 2018; Bernstein et al., 2018). A direct analysis of Adam's implicit bias remains challenging due to its adaptive second-moment estimation. For theoretical clarity, we consider gradient descent without momentum. For Muon, we analyze its idealized form where the orthogonalization is exact; in practice, Muon approximates this via Newton-Schulz iteration.

Consider learning a matrix $W \in \mathbb{R}^{m \times n}$ to satisfy $Wx = y$ for a given $x \in \mathbb{R}^n$ and $y \in \mathbb{R}^m$. The loss function is

$$L(W) = \frac{1}{2}\|Wx - y\|_2^2,$$

with gradient $\nabla_W L = (Wx - y)x^\top$. The initialization is chosen as $W_0 = 0$.

We first prove the following lemma:

**Lemma D.1.** *Let $\{\eta_t\}_{t \geq 0}$ be a sequence of step sizes satisfying:*

*(i) $\eta_t > 0$ for all t,*

*(ii) $\lim_{t \to \infty} \eta_t = 0$,*

*(iii) $\sum_{t=0}^{\infty} \eta_t = \infty$.*

*Then any sequence $\{d_t\}_{t \geq 0}$ defined by the recurrence*

$$d_{t+1} = d_t - \eta_t \operatorname{sign}(d_t)$$

*satisfies $\lim_{t \to \infty} d_t = 0$ for any initial value $d_0 \in \mathbb{R}$.*

*Proof.* Let $r_t := |d_t| \geq 0$. We first analyze the recurrence for $r_t$.

If $d_t = 0$, then $d_{t+1} = 0$ and $r_{t+1} = 0$.

If $d_t \neq 0$, then

$$r_{t+1} = |d_t - \eta_t \operatorname{sign}(d_t)| = ||d_t| - \eta_t| = |r_t - \eta_t|.$$

In either case, $r_{t+1} \leq \max\{r_t, \eta_t\}$.

Since $\lim_{t \to \infty} \eta_t = 0$, for any $\varepsilon > 0$, there exists $T$ such that $\eta_t < \varepsilon$ for all $t \geq T$.

We claim there exists $t_\varepsilon \geq T$ such that $r_{t_\varepsilon} < \varepsilon$.

We prove it by contradiction. Suppose that $r_t \geq \varepsilon$ for all $t \geq T$. Since $\eta_t < \varepsilon \leq r_t$ for $t \geq T$, we have $r_{t+1} = r_t - \eta_t$. Thus

$$r_t = r_T - \sum_{k=T}^{t-1} \eta_k.$$

Since $\sum_{k=T}^{\infty} \eta_k = \infty$, the right-hand side becomes negative for sufficiently large $t$, contradicting $r_t \geq 0$.

Once $r_t < \varepsilon$ for some $t \geq T$, we have

$$r_{t+1} = |r_t - \eta_t| \leq \max\{r_t, \eta_t\} < \varepsilon,$$

since both $r_t < \varepsilon$ and $\eta_t < \varepsilon$. By induction, $r_s < \varepsilon$ for all $s \geq t_\varepsilon$.

Since $\varepsilon > 0$ is arbitrary, we conclude $\lim_{t \to \infty} r_t = 0$, i.e., $\lim_{t \to \infty} d_t = 0$. $\qquad\square$

### D.1. Sign Gradient Descent

**Theorem D.2** (Implicit Bias of SignGD, Restatement of Theorem 3.1). *Let $x \neq 0$. Consider the SignGD iteration from $W_0 = 0$:*

$$W_{t+1} = W_t - \eta_t \operatorname{sign}(\nabla_W L(W_t)),$$

*where step sizes satisfy conditions (i)–(iii) of Lemma D.1. Then*

$$W_t \to W^* = \frac{y \cdot \operatorname{sign}(x)^\top}{\|x\|_1},$$

*and $W^* x = y$. Moreover, $W^*$ achieves the minimum max-norm among all solutions to $W x = y$:*

$$\|W^*\|_{\max} := \max_{i,j} |W_{ij}^*| = \frac{\|y\|_\infty}{\|x\|_1} = \min_{W: Wx=y} \|W\|_{\max}.$$

*Proof.* For any matrix $W$, the gradient is $\nabla_W L(W) = (Wx - y)x^\top$. The $(i,j)$-entry is

$$[\nabla_W L(W)]_{ij} = (Wx - y)_i \cdot x_j,$$

thus

$$\operatorname{sign}(\nabla_W L(W)) = \operatorname{sign}(Wx - y) \cdot \operatorname{sign}(x)^\top.$$

We then prove by induction that $W_t = w_t \cdot \operatorname{sign}(x)^\top$ for some $w_t \in \mathbb{R}^m$.

*Base case*: $W_0 = 0 = 0 \cdot \operatorname{sign}(x)^\top$ with $w_0 = 0$.

*Inductive step*: Suppose $W_t = w_t \cdot \operatorname{sign}(x)^\top$. Then

$$W_t x = w_t(\operatorname{sign}(x)^\top x) = w_t \|x\|_1,$$

and

$$\operatorname{sign}(\nabla_W L(W_t)) = \operatorname{sign}(w_t \|x\|_1 - y) \cdot \operatorname{sign}(x)^\top.$$

The update gives

$$W_{t+1} = w_t \operatorname{sign}(x)^\top - \eta_t \operatorname{sign}(w_t \|x\|_1 - y) \operatorname{sign}(x)^\top = w_{t+1} \operatorname{sign}(x)^\top,$$

where $w_{t+1} = w_t - \eta_t \operatorname{sign}(w_t \|x\|_1 - y)$.

Let $s := \|x\|_1 > 0$ and $w^* := \frac{y}{s}$. Define the error $d_t := w_t - w^*$. Then

$$(d_{t+1})_i = (d_t)_i - \eta_t \operatorname{sign}(s(w_t)_i - y_i) = (d_t)_i - \eta_t \operatorname{sign}((d_t)_i).$$

By Lemma D.1, $\lim_{t \to \infty} (d_t)_i = 0$ for each $i$. Thus $w_t \to w^* = \frac{y}{\|x\|_1}$, and

$$W_t = w_t \operatorname{sign}(x)^\top \to \frac{y \operatorname{sign}(x)^\top}{\|x\|_1} = W^*,$$

which is the solution to the problem given:

$$W^* x = \frac{y\,\mathrm{sign}(x)^\top x}{\|x\|_1} = \frac{y\|x\|_1}{\|x\|_1} = y.$$

Furthermore, we have $\|W^*\|_{\max} = \max_{i,j} |W^*_{ij}| = \frac{\|y\|_\infty}{\|x\|_1}$. For any $W$ satisfying $W x = y$, consider row $i$ where $|y_i| = \|y\|_\infty$. By Hölder's inequality:

$$\|y\|_\infty = |y_i| = |W_i^\top x| \le \|W_i\|_\infty \|x\|_1 \le \|W\|_{\max} \|x\|_1.$$

Thus $\|W\|_{\max} \ge \frac{\|y\|_\infty}{\|x\|_1} = \|W^*\|_{\max}$, so $W^*$ achieves the lower bound. $\qquad\square$

### D.2. Muon

**Theorem D.3** (Implicit Bias of Muon, Restatement of Theorem 3.2). *Let $x \ne 0$ and $y \ne 0$. Define $\mathrm{ortho}(G)$ as the orthogonal factor from the polar decomposition of $G$: if $G = U\Sigma V^\top$ is the compact SVD (where $U$ and $V$ have orthonormal columns), then $\mathrm{ortho}(G) = UV^\top$; we set $\mathrm{ortho}(0) = 0$. Consider the Muon iteration from $W_0 = 0$:*

$$W_{t+1} = W_t - \eta_t \, \mathrm{ortho}(\nabla_W L(W_t)),$$

*where step sizes satisfy conditions (i)–(iii) of Lemma D.1. Then*

$$W_t \to W^* = \frac{yx^\top}{\|x\|_2^2}.$$

*Moreover, $W^*$ achieves the minimum spectral norm among all solutions to $W x = y$:*

$$\|W^*\|_2 = \frac{\|y\|_2}{\|x\|_2} = \min_{W:Wx=y} \|W\|_2.$$

*Proof.* For any matrix $W_t$, the gradient is $\nabla_W L(W_t) = r_t x^\top$ where $r_t := W_t x - y$. When $r_t \ne 0$, this is a rank-1 matrix. For any rank-1 matrix $uv^\top$ with $u, v \ne 0$:

$$\mathrm{ortho}(uv^\top) = \frac{u}{\|u\|_2} \cdot \frac{v^\top}{\|v\|_2}.$$

Thus, when $r_t \ne 0$:

$$\mathrm{ortho}(\nabla_W L(W_t)) = \frac{r_t}{\|r_t\|_2} \cdot \frac{x^\top}{\|x\|_2}.$$

We prove by induction that $W_t = \alpha_t \cdot \frac{yx^\top}{\|x\|_2^2}$ for some scalar $\alpha_t$.

*Base case*: $W_0 = 0$ corresponds to $\alpha_0 = 0$.

*Inductive step*: Suppose the claim holds for $t$. Then

$$W_t x = \alpha_t y, \quad r_t = W_t x - y = (\alpha_t - 1) y.$$

If $\alpha_t \ne 1$ (so $r_t \ne 0$):

$$\frac{r_t}{\|r_t\|_2} = \mathrm{sign}(\alpha_t - 1) \frac{y}{\|y\|_2}.$$

The update gives

$$W_{t+1} = W_t - \eta_t \,\mathrm{sign}(\alpha_t - 1) \frac{y}{\|y\|_2} \cdot \frac{x^\top}{\|x\|_2} = (\alpha_t - \beta_t \,\mathrm{sign}(\alpha_t - 1)) \frac{yx^\top}{\|x\|_2^2},$$

where $\beta_t := \eta_t \frac{\|x\|_2}{\|y\|_2} > 0$. Thus $\alpha_{t+1} = \alpha_t - \beta_t \,\mathrm{sign}(\alpha_t - 1)$.

Let $d_t := \alpha_t - 1$. Then

$$d_{t+1} = d_t - \beta_t \operatorname{sign}(d_t).$$

Since $\lim_{t\to\infty} \eta_t = 0$ and $\sum_t \eta_t = \infty$, we have $\lim_{t\to\infty} \beta_t = 0$ and $\sum_t \beta_t = \infty$. By Lemma D.1, $\lim_{t\to\infty} d_t = 0$, i.e., $\lim_{t\to\infty} \alpha_t = 1$. Therefore

$$W_t = \alpha_t \frac{yx^\top}{\|x\|_2^2} \to \frac{yx^\top}{\|x\|_2^2} = W^*.$$

Lastly, for any $W$ satisfying $Wx = y$:

$$\|W\|_2 = \max_{\|v\|_2=1} \|Wv\|_2 \geq \frac{\|Wx\|_2}{\|x\|_2} = \frac{\|y\|_2}{\|x\|_2}.$$

Since $W^*$ has rank 1:

$$\|W^*\|_2 = \frac{\|y\|_2 \cdot \|x\|_2}{\|x\|_2^2} = \frac{\|y\|_2}{\|x\|_2}.$$

Thus, $W^*$ achieves the lower bound. $\qquad\square$

### D.3. LoRA Mitigates Mismatch: Theoretical Analysis

Throughout this subsection, we continue to use SignGD as a proxy for Adam and idealized Muon (exact orthogonalization and no momentum), exactly as in the preceding sections. Step sizes are assumed to satisfy the conditions of Lemma D.1. We assume $r_0 \neq 0$ whenever ratios or nonempty mismatch intervals are discussed.

Consider a pretrained matrix $W_0 \in \mathbb{R}^{m \times n}$ and a fine-tuning example $(z, b)$ with $z \neq 0$. The fine-tuning loss is

$$L_{\text{ft}}(W) = \frac{1}{2}\|Wz - b\|_2^2.$$

Define the correction

$$\Delta := W - W_0, \qquad r_0 := b - W_0 z.$$

Then

$$L_{\text{ft}}(W_0 + \Delta) = \frac{1}{2}\|\Delta z - r_0\|_2^2.$$

Thus, fine-tuning from $W_0$ is equivalent to learning from zero a correction matrix $\Delta$ that fits the residual $r_0$.

**Proposition D.4 (Continuation from a pretrained weight).** Under the above setup, SignGD initialized at $W_0$ converges to

$$W_{\text{s}}^\star(W_0) = W_0 + \Delta_{\text{s}}^\star, \qquad \Delta_{\text{s}}^\star = \frac{r_0 \operatorname{sign}(z)^\top}{\|z\|_1},$$

while the idealized Muon initialized at $W_0$ converges to

$$W_\mu^\star(W_0) = W_0 + \Delta_\mu^\star, \qquad \Delta_\mu^\star = \frac{r_0 z^\top}{\|z\|_2^2}.$$

Moreover,

$$\Delta_{\text{s}}^\star \in \arg\min_{\Delta:\,\Delta z = r_0} \|\Delta\|_{\max}, \qquad \Delta_\mu^\star \in \arg\min_{\Delta:\,\Delta z = r_0} \|\Delta\|_2.$$

*Proof.* The gradient of $\Delta \mapsto \frac{1}{2}\|\Delta z - r_0\|_2^2$ is $\nabla_\Delta L = (\Delta z - r_0)z^\top$, which is exactly the same form as in Theorems D.2 and D.3, with $x$ replaced by $z$ and $y$ replaced by $r_0$. Since $\Delta_0 = 0$, the claims follow immediately. $\square$

**Proposition D.5 (Budgeted fine-tuning in native geometries).** For $\rho \geq 0$, define the budgeted fine-tuning objectives

$$\mathcal{E}_{\max}(\rho) := \min_{\|\Delta\|_{\max} \leq \rho} \frac{1}{2} \|\Delta z - r_0\|_2^2, \qquad \mathcal{E}_2(\rho) := \min_{\|\Delta\|_2 \leq \rho} \frac{1}{2} \|\Delta z - r_0\|_2^2.$$

Then

$$\mathcal{E}_{\max}(\rho) = \frac{1}{2} \sum_{i=1}^{m} \left( |(r_0)_i| - \rho \|z\|_1 \right)_+^2,$$

and

$$\mathcal{E}_2(\rho) = \frac{1}{2} \left( \|r_0\|_2 - \rho \|z\|_2 \right)_+^2.$$

The smallest budgets that permit exact fit are

$$\rho_{\mathrm{A}}^\star := \frac{\|r_0\|_\infty}{\|z\|_1}, \qquad \rho_\mu^\star := \frac{\|r_0\|_2}{\|z\|_2}.$$

Thus, in the Adam/SignGD-native max-norm geometry, the matched optimizer reaches exact fit with the smallest max-norm budget; in the Muon-native spectral geometry, the matched optimizer reaches exact fit with the smallest spectral budget.

*Proof.* Under the constraint $\|\Delta\|_{\max} \leq \rho$, the rows decouple: each row $\delta_i$ satisfies $|\delta_i^\top z| \leq \rho \|z\|_1$, and any scalar in $[-\rho\|z\|_1, \rho\|z\|_1]$ is attained by $\delta_i = t_i \operatorname{sign}(z)/\|z\|_1$. Each term is minimized by projecting $(r_0)_i$ onto this interval. For the spectral problem, let $u := \Delta z$. If $\|\Delta\|_2 \leq \rho$, then $\|u\|_2 \leq \rho\|z\|_2$, and conversely $\Delta = uz^\top/\|z\|_2^2$ achieves any such $u$. The result is the Euclidean projection of $r_0$ onto the ball of radius $\rho\|z\|_2$. $\square$

**Corollary D.6 (Matched exact-fit thresholds under native budgets).** Let $\Delta_{\mathrm{s}}^\star$ and $\Delta_\mu^\star$ be the exact-fit corrections from Proposition D.4. Define

$$\tilde{\rho}_{\mu|\max}^\star := \|\Delta_\mu^\star\|_{\max}, \qquad \tilde{\rho}_{\mathrm{s}|2}^\star := \|\Delta_{\mathrm{s}}^\star\|_2.$$

Then

$$\tilde{\rho}_{\mu|\max}^\star = \frac{\|r_0\|_\infty \|z\|_\infty}{\|z\|_2^2} \geq \frac{\|r_0\|_\infty}{\|z\|_1} = \rho_{\mathrm{A}}^\star,$$

and

$$\tilde{\rho}_{\mathrm{s}|2}^\star = \frac{\sqrt{\|z\|_0}\,\|r_0\|_2}{\|z\|_1} \geq \frac{\|r_0\|_2}{\|z\|_2} = \rho_\mu^\star.$$

In particular, whenever either inequality is strict, there exists a nonempty native-budget interval in which the matched exact-fit correction is already feasible while the mismatched exact-fit correction is not.

*Proof.* The inequalities are equivalent to $\|z\|_2^2 \leq \|z\|_1 \|z\|_\infty$ and $\|z\|_1 \leq \sqrt{\|z\|_0}\,\|z\|_2$, respectively. $\square$

**Corollary D.7 (Old-task damage of exact-fit fine-tuning).** Suppose $W_0 x = y$. Then for any fine-tuning correction $\Delta$,

$$L_{\mathrm{old}}(W_0 + \Delta) = \frac{1}{2} \|\Delta x\|_2^2 \leq \frac{m}{2} \|\Delta\|_{\max}^2 \|x\|_1^2,$$

and also $L_{\mathrm{old}}(W_0 + \Delta) \leq \frac{1}{2} \|\Delta\|_2^2 \|x\|_2^2$. Therefore, among all exact-fit fine-tuning solutions $Wz = b$, matched SignGD minimizes the first upper bound (max-norm), while matched Muon minimizes the second (spectral norm).

*Proof.* Since $W_0 x = y$, $L_{\mathrm{old}}(W_0 + \Delta) = \frac{1}{2} \|\Delta x\|_2^2$. The bounds follow from $|(\Delta x)_i| \leq \|\Delta\|_{\max} \|x\|_1$ and $\|\Delta x\|_2 \leq \|\Delta\|_2 \|x\|_2$. The optimality claims follow from Proposition D.4. $\square$

## D.4. A Fixed-Subspace LoRA Surrogate

In the one-sample model above, standard LoRA with both factors trainable is too expressive: both $\Delta_{\mathrm{s}}^\star$ and $\Delta_\mu^\star$ are rank-one, so rank-one LoRA can represent them exactly. To isolate the effect of constraining updates to a low-dimensional adapter geometry, we consider the fixed-subspace surrogate

$$W = W_0 + BA,$$

where $A \in \mathbb{R}^{r \times n}$ is fixed, $B \in \mathbb{R}^{m \times r}$ is trainable, $B_0 = 0$, and $Az \neq 0$.

The fine-tuning loss becomes

$$L(B) = \frac{1}{2} \|BAz - r_0\|_2^2.$$

Let $u := Az \in \mathbb{R}^r$. Then $L(B) = \frac{1}{2}\|Bu - r_0\|_2^2$, which is of the same form as the residual problem above, but with input $u$ instead of $z$.

**Proposition D.8 (Budgeted fine-tuning under a fixed-subspace LoRA surrogate).** Under the fixed-subspace surrogate, SignGD and idealized Muon converge to

$$B_{\mathrm{s}}^{\star} = \frac{r_0 \operatorname{sign}(u)^{\top}}{\|u\|_1}, \qquad B_{\mu}^{\star} = \frac{r_0 u^{\top}}{\|u\|_2^2}.$$

The corresponding exact-fit adapter-budget thresholds are

$$\rho_{\mathrm{A,LoRA}}^{\star} = \frac{\|r_0\|_{\infty}}{\|Az\|_1}, \qquad \rho_{\mu,\mathrm{LoRA}}^{\star} = \frac{\|r_0\|_2}{\|Az\|_2}.$$

The mismatched exact-fit adapter budgets are

$$\tilde{\rho}_{\mu|\max,\mathrm{LoRA}}^{\star} := \|B_{\mu}^{\star}\|_{\max} = \frac{\|r_0\|_{\infty}\|Az\|_{\infty}}{\|Az\|_2^2}, \qquad \tilde{\rho}_{\mathrm{s}|2,\mathrm{LoRA}}^{\star} := \|B_{\mathrm{s}}^{\star}\|_2 = \frac{\sqrt{\|Az\|_0}\,\|r_0\|_2}{\|Az\|_1}.$$

Hence, the LoRA mismatch inflation factors satisfy

$$\frac{\tilde{\rho}_{\mu|\max,\mathrm{LoRA}}^{\star}}{\rho_{\mathrm{A,LoRA}}^{\star}} = \frac{\|Az\|_1\|Az\|_{\infty}}{\|Az\|_2^2} \leq \|Az\|_0 \leq r,$$

and

$$\frac{\tilde{\rho}_{\mathrm{s}|2,\mathrm{LoRA}}^{\star}}{\rho_{\mu,\mathrm{LoRA}}^{\star}} = \frac{\sqrt{\|Az\|_0}\,\|Az\|_2}{\|Az\|_1} \leq \sqrt{\|Az\|_0} \leq \sqrt{r}.$$

If $r = 1$, then $B_{\mathrm{s}}^{\star} = B_{\mu}^{\star}$, so in each native geometry the matched and mismatched exact-fit adapter budgets coincide and the mismatch inflation factor equals one. If $A = I$, they reduce to the full fine-tuning thresholds from Proposition D.5 and Corollary D.6.

In addition, if $W_0 x = y$, then

$$L_{\mathrm{old}}(W_0 + B_{\mathrm{s}}^{\star} A) = \frac{1}{2}\|r_0\|_2^2 \frac{\langle \operatorname{sign}(Az), Ax \rangle^2}{\|Az\|_1^2},$$

and

$$L_{\mathrm{old}}(W_0 + B_{\mu}^{\star} A) = \frac{1}{2}\|r_0\|_2^2 \frac{\langle Az, Ax \rangle^2}{\|Az\|_2^4}.$$

*Proof.* Apply Proposition D.4 and Proposition D.5 to the optimization problem in $B$, with input $u = Az$. The inflation factor bounds follow from $\|u\|_1 \leq \|u\|_0\|u\|_{\infty}$, $\|u\|_{\infty}^2 \leq \|u\|_2^2$, and $\|u\|_1 \geq \|u\|_2$, combined with $\|u\|_0 \leq r$. $\square$

**Summary.** Propositions D.4–D.8 formalize the mismatch mechanism in this simplified setting. Fine-tuning from a pretrained model is a residual-fitting problem in which the relevant object is the correction matrix $\Delta$. Under a fixed native update budget, the matched optimizer reaches exact fit with the smallest geometry-aligned budget; under exact fit, old-task damage is controlled by the same native geometry. The fixed-subspace LoRA surrogate replaces the full geometry $z$ by the effective adapter geometry $Az$, so both the exact-fit thresholds and the old-task damage formulas are governed by the compressed adapter geometry. In particular, the worst-case LoRA mismatch inflation is at most $r$ in Adam-native max geometry and at most $\sqrt{r}$ in Muon-native spectral geometry, while $r = 1$ collapses the gap and $A = I$ recovers the full fine-tuning formulas.

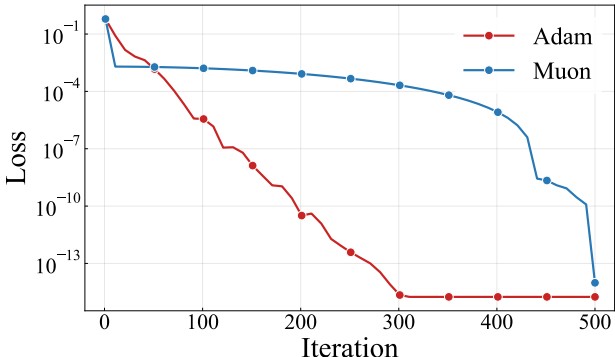

*Figure 9.* Loss curves for the implicit bias experiment. Both Adam and Muon converge to near-zero loss, finding valid solutions to the underdetermined linear system.

**Numerical Verification.** Figure 9 shows the loss curves for the numerical verification experiment in Figure 2 (left). Both optimizers successfully minimize the loss to near-zero, confirming that they both find solutions to the underdetermined system, albeit with different implicit biases.

## E. Experimental Details

### E.1. Natural Language Understanding

**Training Details.** We use T5-Base. Models are trained for 5 epochs on MRPC and CoLA, and 3 epochs on SST-2, QNLI, and MNLI, with a batch size of 64 and a sequence length of 128. For practical evaluation, we use the original validation set as the test set and hold out 10% of the training data for validation. We perform 5 evaluations during training and select the best checkpoint based on validation performance; in practice, the final checkpoint is consistently the best across all experiments. All experiments use FP32 precision.

**Optimizer Settings.** For Adam, we use $\beta_1 = 0.9$, $\beta_2 = 0.999$, $\epsilon = $ 1e-8, and weight decay of 0, with a warmup ratio of 0.03 and cosine learning rate decay. For Muon, we use momentum $= 0.95$ with Nesterov momentum enabled, Newton-Schulz iteration with 5 steps (NS5) computed in BF16 precision, no weight decay, and shape-dependent learning rate scaling (see Appendix B). We also evaluate Muon with Polar Express (PE) coefficients (Amsel et al., 2026), using their default settings with lower bound $\ell = $ 1e-3, 10 iterations, safety factor 1e-2, and cushion factor 0.02.

Following the standard Muon implementation (Jordan et al., 2024), embedding layers and the language modeling (LM) head are optimized with Adam while other parameters use Muon. Since T5 uses simplified layer normalization without bias terms, there are no 1D parameters. For full fine-tuning, the embedding layer and LM head use Adam, with $\beta_1 = 0.9$, $\beta_2 = 0.95$, $\epsilon = $ 1e-8, and no weight decay, while all other parameters use Muon. For LoRA fine-tuning, since LoRA is only applied to linear layers, all trainable parameters are optimized entirely with Muon.

**Learning Rate Selection.** We perform a learning rate sweep over {1e-5, 5e-5, 1e-4, 5e-4, 1e-3, 2e-3, 5e-3, 1e-2} for each method on each dataset. Table 9 shows the selected learning rates.

*Table 9.* Selected learning rates for NLU experiments.

| Method | CoLA | MNLI | MRPC | QNLI | SST-2 |
|---|---|---|---|---|---|
| Full-Adam | 1e-4 | 1e-4 | 1e-4 | 5e-5 | 1e-4 |
| Full-Muon | 5e-4 | 1e-4 | 5e-4 | 1e-4 | 1e-4 |
| Full-Muon-PE | 1e-3 | 1e-4 | 5e-4 | 1e-4 | 1e-4 |
| LoRA-Adam | 1e-3 | 1e-3 | 2e-3 | 5e-4 | 5e-4 |
| LoRA-Muon | 2e-3 | 1e-3 | 2e-3 | 5e-4 | 1e-3 |
| LoRA-Muon-PE | 2e-3 | 1e-3 | 2e-3 | 5e-4 | 1e-3 |

### E.2. Natural Language Generation

**Training Details.** We use Llama 2-7B. Models are trained for 1 epoch with a batch size of 32 and a sequence length of 1024. The backbone model uses BF16 precision, while LoRA's A and B matrices use FP32 precision following the PEFT (Mangrulkar et al., 2022) implementation. We apply LoRA with rank $r = 8$ and $\alpha = 16$ to all linear layers except for the embeddings and the language model head. We found that the final checkpoint consistently achieves the best performance, so we evaluate on the final model.

**Optimizer Settings.** For Adam, we use $\beta_1 = 0.9$, $\beta_2 = 0.999$, $\epsilon = $ 1e-8, and no weight decay, with a warmup ratio of 0.03 and cosine learning rate decay. For Muon, we use the same settings as in the NLU experiments (Section E.1). Following the standard Muon implementation, 1D parameters (biases and layer normalization weights), embedding layers, and the LM head are optimized with Adam while other parameters use Muon. For LoRA fine-tuning, all trainable parameters are optimized with Muon.

**Evaluation.** All evaluations are conducted using lm-evaluation-harness (Gao et al., 2024). For HumanEval and GSM8K, we modify the prompts to match the instruction tuning format and use greedy decoding with a maximum generation length of 1024 tokens. For commonsense benchmarks, we use the default prompts.

**Learning Rate Selection.** We perform a learning rate sweep over {1e-6, 5e-6, 8e-6, 1e-5, 2e-5, 5e-5, 1e-4, 5e-4} for each method on each task. Table 10 shows the selected learning rates.

*Table 10.* Selected learning rates for NLG experiments.

| Method | Math | Code | Commonsense |
|---|---|---|---|
| Full-Adam | 1e-5 | 1e-5 | 8e-6 |
| Full-Muon | 5e-5 | 5e-5 | 2e-5 |
| Full-Muon-PE | 5e-5 | 5e-5 | 2e-5 |
| LoRA-Adam | 5e-4 | 5e-4 | 2e-4 |
| LoRA-Muon | 5e-4 | 5e-4 | 1e-4 |
| LoRA-Muon-PE | 5e-4 | 5e-4 | 1e-4 |

**Larger-Scale Experiment: Llama 2-13B.** We also extended our LoRA experiments to Llama 2-13B on CodeFeedback using the same settings as above. We swept learning rates in {1e-5, 3e-5, 5e-5, 7e-5, 1e-4, 3e-4, 5e-4, 7e-4, 9e-4} and report HumanEval Pass@1 averaged over 3 seeds (best learning rate = 7e-4 for both methods):

- LoRA-Adam: $33.17_{\pm 1.17}\%$

- LoRA-Muon: $34.76_{\pm 2.44}\%$

LoRA-Muon performs comparably to LoRA-Adam at this larger scale, consistent with the Llama 2-7B results in Table 3. For full fine-tuning at 13B, memory constraints prevent standard DDP (which we use to ensure fair comparison with the original Muon implementation); we leave this to future work as Muon's compatibility with memory-reduction frameworks improves.

### E.3. Image Classification

**Training Details.** We use CLIP ViT-B/32 and freeze the text tower. For each dataset, we use templates and class names from CLIP-Benchmark (Cherti & Beaumont, 2025) to build a template-ensemble text classifier, and the resulting text features (and logit scale) are cached and treated as constants. We then optimize the vision branch with a cross-entropy objective over the induced image–text logits. All models are trained for 40 epochs with a batch size of 256 under BF16 mixed precision. We use cosine learning rate decay with a warmup ratio of 0.03, weight decay of 0.1, and gradient clipping with a max norm of 1.0. We use native train/test splits from dataset sources, select the final checkpoint, and report its test accuracy.

**Optimizer Settings.** For Adam, we use $\beta_1 = 0.9$, $\beta_2 = 0.999$, $\epsilon = $ 1e-8, and weight decay 0.1. For Muon, we use momentum 0.95 with Nesterov momentum enabled, Newton-Schulz iteration with 5 steps computed in BF16, and the same weight decay 0.1. Following standard Muon practice, matrix-shaped parameters (including the CLIP visual projection) are optimized with Muon, while embedding-like and other non-matrix parameters (including 1D parameters and patch/class/position embeddings) are optimized with an Adam sub-optimizer. For LoRA fine-tuning, we inject adapters into the q_proj/v_proj/visual_projection layers with rank $r = 8$, $\alpha = 16$, and dropout 0.0; all LoRA adapters use Muon regardless of their parent layer. Muon-PE uses PE coefficients in the Newton-Schulz backend.

**Learning Rate Selection.** For full fine-tuning, we perform a learning rate sweep over {5e-6, 1e-5, 5e-5, 1e-4}. For LoRA, we sweep {5e-5, 1e-4, 5e-4, 1e-3}. Table 11 shows the selected learning rates.

*Table 11.* Selected learning rates for image classification experiments.

| Method | StanfordCars | DTD | GTSRB | RESISC45 | SUN397 | SVHN |
|---|---|---|---|---|---|---|
| Full-Adam | 1e-5 | 1e-5 | 1e-5 | 1e-5 | 1e-5 | 1e-5 |
| Full-Muon | 5e-5 | 1e-5 | 1e-5 | 1e-5 | 1e-5 | 5e-5 |
| Full-Muon-PE | 5e-5 | 1e-5 | 1e-5 | 1e-5 | 1e-5 | 5e-5 |
| LoRA-Adam | 1e-3 | 1e-3 | 1e-3 | 1e-3 | 5e-4 | 1e-3 |
| LoRA-Muon | 1e-3 | 1e-3 | 5e-4 | 1e-3 | 5e-4 | 1e-3 |
| LoRA-Muon-PE | 1e-3 | 1e-3 | 1e-3 | 1e-3 | 5e-4 | 1e-3 |

**Wall-Clock Time.** Tables 12 and 13 report training time for Llama 2-7B and CLIP ViT-B/32 experiments, respectively.

*Table 12.* Wall-clock time (hours) for Llama 2-7B fine-tuning on 8×AMD MI210 GPUs, averaged over 3 seeds. For full fine-tuning, Adam uses DeepSpeed ZeRO-2 (required to fit in memory), while Muon's memory-efficient design enables standard DDP, accounting for the larger gap. Parentheses show relative time vs. Adam (red = slower, green = faster).

| | | Math | Code | Commonsense |
|---|---|---|---|---|
| Full | Adam | 2.52h | 3.43h | 1.88h |
| | Muon | 7.25h (2.9×) | 8.14h (2.4×) | 4.39h (2.3×) |
| | Muon-PE | 7.26h (2.9×) | 8.15h (2.4×) | 4.39h (2.3×) |
| LoRA | Adam | 1.02h | 1.72h | 0.96h |
| | Muon | 1.27h (1.2×) | 1.97h (1.1×) | 1.09h (1.1×) |
| | Muon-PE | 1.28h (1.3×) | 1.99h (1.2×) | 1.10h (1.1×) |

*Table 13.* Wall-clock time (minutes) for CLIP ViT-B/32 fine-tuning on 1×A100 GPU, averaged over 3 seeds. Parentheses show relative time vs. Adam (red = slower, green = faster).

| | | DTD | GTSRB | RESISC | Cars | SUN397 | SVHN |
|---|---|---|---|---|---|---|---|
| Full | Adam | 3.8 | 17.9 | 11.6 | 14.4 | 72.9 | 38.6 |
| | Muon | 4.0 (1.1×) | 20.7 (1.2×) | 13.9 (1.2×) | 14.7 (1.0×) | 73.5 (1.0×) | 46.7 (1.2×) |
| | Muon-PE | 4.0 (1.1×) | 20.5 (1.1×) | 14.2 (1.2×) | 14.4 (1.0×) | 68.3 (0.9×) | 46.1 (1.2×) |
| LoRA | Adam | 3.6 | 16.2 | 10.5 | 14.4 | 75.0 | 33.6 |
| | Muon | 3.8 (1.1×) | 17.6 (1.1×) | 11.5 (1.1×) | 14.7 (1.0×) | 75.1 (1.0×) | 37.3 (1.1×) |
| | Muon-PE | 3.9 (1.1×) | 17.5 (1.1×) | 12.2 (1.2×) | 14.0 (1.0×) | 69.1 (0.9×) | 37.2 (1.1×) |

### E.4. LoRA Rank Study

This section provides details for the LoRA rank study in Section 4.4. For each rank $r \in \{2, 4, 8, 16, 32, 64, 128, 256, 512\}$, we set $\alpha = 2r$ and perform a learning rate sweep. For MetaMath and CodeFeedback, the sweep covers {1e-5, 3e-5, 5e-5, 7e-5, 1e-4, 3e-4, 5e-4, 7e-4, 1e-3}, and all other hyperparameters match the NLG experiments (Section E.2). For StanfordCars, the sweep covers {1e-4, 3e-4, 5e-4, 7e-4, 1e-3, 3e-3, 5e-3, 7e-3, 1e-2}, and all other hyperparameters match the image classification experiments (Section E.3). Tables 14, 15, and 16 show the selected learning rates for each rank.

*Table 14.* Selected learning rates for LoRA rank study on MetaMath.

| Rank | LoRA-Muon | LoRA-Adam |
|------|-----------|-----------|
| $r = 2$ | 7e-4 | 1e-3 |
| $r = 4$ | 5e-4 | 1e-3 |
| $r = 8$ | 5e-4 | 5e-4 |
| $r = 16$ | 3e-4 | 5e-4 |
| $r = 32$ | 3e-4 | 3e-4 |
| $r = 64$ | 3e-4 | 1e-4 |
| $r = 128$ | 1e-4 | 1e-4 |
| $r = 256$ | 1e-4 | 1e-4 |
| $r = 512$ | 7e-5 | 7e-5 |

*Table 15.* Selected learning rates for LoRA rank study on Code-Feedback.

| Rank | LoRA-Muon | LoRA-Adam |
|------|-----------|-----------|
| $r = 2$ | 5e-4 | 1e-3 |
| $r = 4$ | 7e-4 | 7e-4 |
| $r = 8$ | 5e-4 | 5e-4 |
| $r = 16$ | 5e-4 | 5e-4 |
| $r = 32$ | 5e-4 | 3e-4 |
| $r = 64$ | 3e-4 | 3e-4 |
| $r = 128$ | 3e-4 | 3e-4 |
| $r = 256$ | 1e-4 | 1e-4 |
| $r = 512$ | 7e-5 | 1e-4 |

### E.5. Catastrophic Forgetting Evaluation

This section provides details for the catastrophic forgetting evaluation in Section 4.5. We use the Llama 2-7B models fine-tuned on MetaMath in Section 4.2 and evaluate them on benchmarks that assess knowledge acquired during pretraining but unrelated to the fine-tuning domain. Following Kotha et al. (2024) and Li et al. (2024a), we exclude benchmarks where performance improves after fine-tuning, as these do not reflect forgetting of pretrained knowledge.

**Commonsense Reasoning (for Math Fine-tuning).** For models fine-tuned on MetaMath, we evaluate on commonsense reasoning benchmarks: ARC-Challenge, ARC-Easy, HellaSwag, OpenBookQA, and PIQA. We exclude WinoGrande and BoolQ as they showed improved performance after fine-tuning. This filtering ensures that the reported metrics genuinely reflect forgetting rather than being confounded by task transfer effects.

**Weight Distance from Pretrained Model.** Table 17 reports the L2 and cosine distance between fine-tuned and pretrained weights for the Llama 2-7B models in Section 4.2, normalized so that Adam = $1.0\times$.

### E.6. LoRA Variants

For the experiments in Section 4.6, we use the same training setup as the NLU experiments (Section E.1). Table 18 shows the selected learning rates for each method.

We also experimented with LoFT (Tastan et al., 2026), but after tuning, it did not outperform LoRA-Adam on our benchmarks. For reference, on GLUE with T5-Base, LoFT achieves an average of 88.83% (CoLA: 82.45$_{\pm0.75}$%, MNLI: 86.14$_{\pm0.07}$%, MRPC: 87.99$_{\pm0.42}$%, QNLI: 93.15$_{\pm0.11}$%, SST-2: 94.42$_{\pm0.24}$%), compared to LoRA-Adam's 88.93%. Among algorithm-modifying methods, we included LoRA-Pro and LoRA-RITE, which outperform LoRA-Adam on this benchmark.

**Variant-Specific Settings.** We use the default settings from each method's official implementation unless otherwise specified.

*Table 16.* Selected learning rates for LoRA rank study on StanfordCars.

| Rank | LoRA-Muon | LoRA-Adam |
|---|---|---|
| $r = 2$ | 1e-2 | 1e-2 |
| $r = 4$ | 5e-3 | 7e-3 |
| $r = 8$ | 3e-3 | 5e-3 |
| $r = 16$ | 3e-3 | 3e-3 |
| $r = 32$ | 1e-3 | 1e-3 |
| $r = 64$ | 1e-3 | 1e-3 |
| $r = 128$ | 7e-4 | 5e-4 |
| $r = 256$ | 3e-4 | 3e-4 |
| $r = 512$ | 3e-4 | 3e-4 |

*Table 17.* Distance from pretrained weights relative to Adam (normalized so Adam = 1.0×). Values >1 indicate the optimizer moves weights farther from the pretrained model than Adam; <1 indicates closer.

| Dataset | Optimizer | Full Fine-Tuning | | LoRA | |
|---|---|---|---|---|---|
| | | L2 | Cos. Dist. | L2 | Cos. Dist. |
| Math | Adam | 1.00× | 1.00× | 1.00× | 1.00× |
| | Muon | 2.37× | 5.61× | 0.83× | 0.69× |
| | Muon-PE | 2.71× | 7.36× | 0.91× | 0.82× |
| Code | Adam | 1.00× | 1.00× | 1.00× | 1.00× |
| | Muon | 2.44× | 5.93× | 0.79× | 0.62× |
| | Muon-PE | 2.62× | 6.86× | 0.80× | 0.65× |
| Commonsense | Adam | 1.00× | 1.00× | 1.00× | 1.00× |
| | Muon | 0.80× | 0.65× | 0.38× | 0.15× |
| | Muon-PE | 0.87× | 0.75× | 0.42× | 0.18× |

- **AdaLoRA**: We set the target average rank to $r = 8$ to match the rank used in other methods.

- **PiSSA**: We use full SVD for initialization.

- **LoRA-One**: We use `stable_gamma=64` and approximate the negative gradient $-G$ using `torch.svd_lowrank` (Ansel et al., 2024) with $q = 512$ and `niter=16`. For gradient estimation, we use a batch size of 1 and 8 iterations.

# F. Additional Results

## F.1. Weight Spectral Analysis

This section provides additional spectral analysis of the attention weights during NanoChat pretraining, complementing Figure 2 (right) in the main text.

Figure 10 shows both the stable rank and SVD entropy of the attention QKV projection weights. For a weight matrix $W$ with singular values $\sigma_1 \geq \sigma_2 \geq \cdots \geq \sigma_r$, we define:

- **Stable rank**: $\mathrm{srank}(W) = \|W\|_F^2 / \|W\|_2^2 = \sum_i \sigma_i^2 / \sigma_1^2$. This measures the effective dimensionality of the weight matrix; a higher stable rank indicates the matrix utilizes more of its capacity rather than being dominated by a few large singular values.

- **SVD entropy**: $H(W) = -\sum_i p_i \log p_i / \log r$, where $p_i = \sigma_i^2 / \sum_j \sigma_j^2$. This quantifies the dispersion of singular values, normalized to $[0, 1]$; a higher entropy indicates a more uniform distribution.

Figure 11 provides a more detailed breakdown, showing the stable rank and SVD entropy separately for query (Q), key (K),

*Table 18.* Selected learning rates for LoRA variants experiments.

| Method | CoLA | MNLI | MRPC | QNLI | SST-2 |
|---|---|---|---|---|---|
| rsLoRA-Adam | 5e-4 | 5e-4 | 2e-3 | 1e-3 | 5e-4 |
| LoRA-One-Adam | 5e-4 | 1e-3 | 1e-3 | 5e-4 | 1e-3 |
| PiSSA-Adam | 5e-4 | 5e-4 | 5e-4 | 1e-4 | 5e-4 |
| rsLoRA-Muon-PE | 1e-3 | 5e-4 | 1e-3 | 5e-4 | 5e-4 |
| LoRA-One-Muon-PE | 2e-3 | 1e-3 | 2e-3 | 1e-3 | 5e-4 |
| PiSSA-Muon-PE | 5e-4 | 5e-4 | 1e-3 | 5e-4 | 5e-4 |
| AdaLoRA-Adam | 5e-3 | 1e-3 | 5e-3 | 1e-2 | 2e-3 |
| LoRA-Pro-Adam | 5e-4 | 5e-4 | 1e-3 | 1e-4 | 1e-4 |
| LoRA-RITE-Adam | 1e-3 | 2e-3 | 1e-3 | 1e-3 | 1e-3 |
| DoRA-Adam | 1e-3 | 5e-4 | 2e-3 | 5e-4 | 2e-3 |

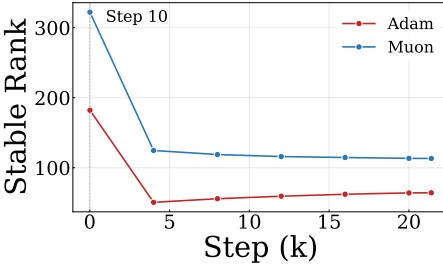 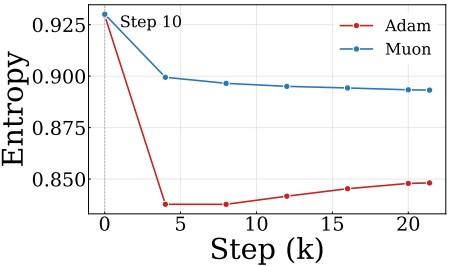

*Figure 10.* Spectral properties of attention QKV projection weights during NanoChat pretraining. **Left:** Stable rank. **Right:** SVD entropy. Muon-trained weights consistently maintain higher stable rank and entropy throughout training, indicating a more distributed spectral structure.

and value (V) projections. The difference between Muon and Adam is consistent across all three projection types, with Muon producing weights that have a higher stable rank and entropy in each case.

### F.2. Spectral Analysis of LoRA Matrices

We analyze the spectral properties of the LoRA $A$ and $B$ matrices during Llama 2-7B fine-tuning, using the same settings as Table 3. Figures 12, 13, and 14 show the stable rank and normalized SVD entropy of LoRA matrices throughout training for attention (Q/K/V/O) and dense layers across all three tasks.

LoRA-Muon consistently produces higher stable rank ($\sim$6–7 vs. $\sim$3–5 for Adam) and entropy ($\sim$0.98–1.0 vs. $\sim$0.80–0.95 for Adam) across all layer types, mirroring the patterns observed in pretrained weights (Section 3.1 and Appendix F.1). This confirms that Muon's implicit bias toward uniform singular value distributions extends to the LoRA matrices.

Notably, LoRA learns on freshly initialized $A$ and $B$ matrices rather than directly modifying pretrained weights. This may help explain why LoRA mitigates mismatch: Muon can freely express its spectral implicit bias on these new matrices, while the pretrained knowledge remains preserved in the frozen base weights. In contrast, full fine-tuning forces Muon to directly alter Adam-shaped weights, causing disruption.

## G. Computational Resources

T5-Base experiments (NLU and LoRA variants) were trained and evaluated on a single AMD Instinct MI210 GPU. Llama 2-7B/13B experiments (NLG, rank study, and catastrophic forgetting) were trained on 8$\times$ AMD Instinct MI210 GPUs and evaluated on 8$\times$ NVIDIA A6000 GPUs. CLIP ViT-B/32 experiments (image classification) were trained and evaluated on a single NVIDIA A100 40GB GPU.

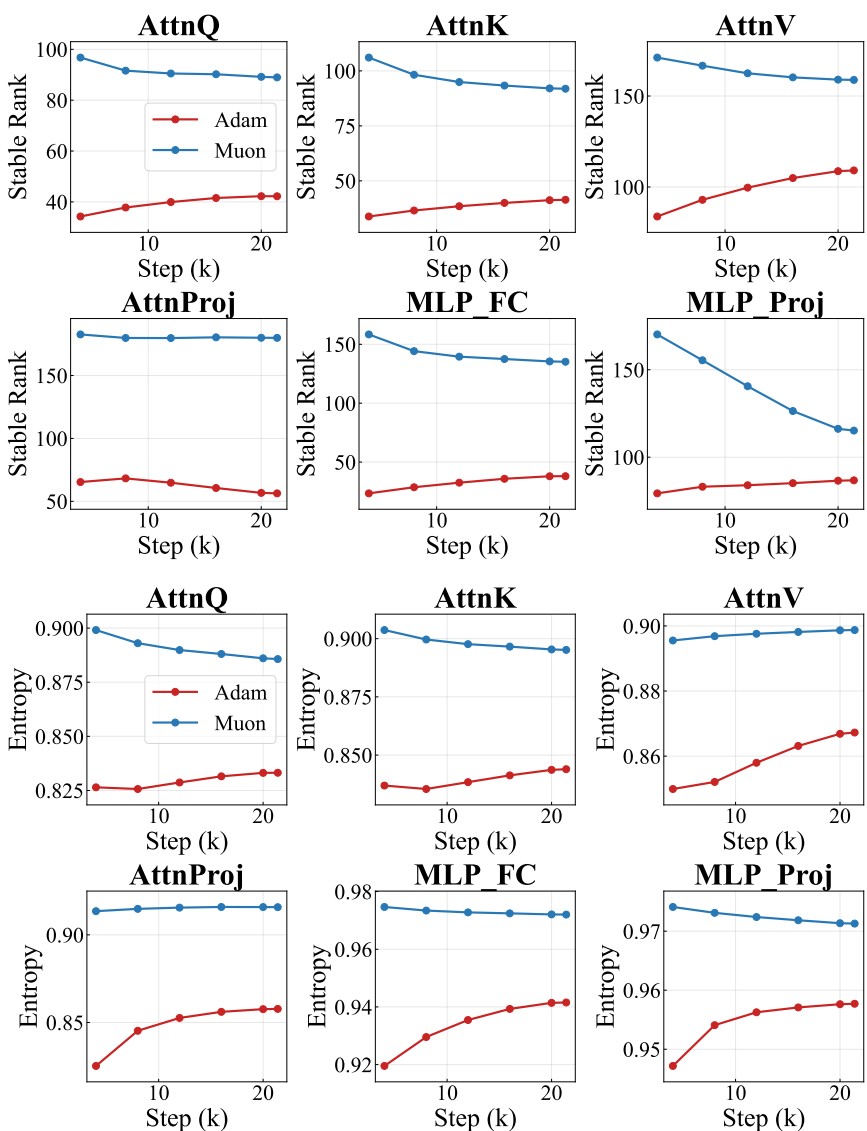

*Figure 11.* Detailed spectral analysis by parameter type. **Top:** Stable rank for Q, K, V projections and MLP layers separately. **Bottom:** SVD entropy for Q, K, V projections and MLP layers separately. The spectral differences between Muon and Adam are consistent across all parameter types.

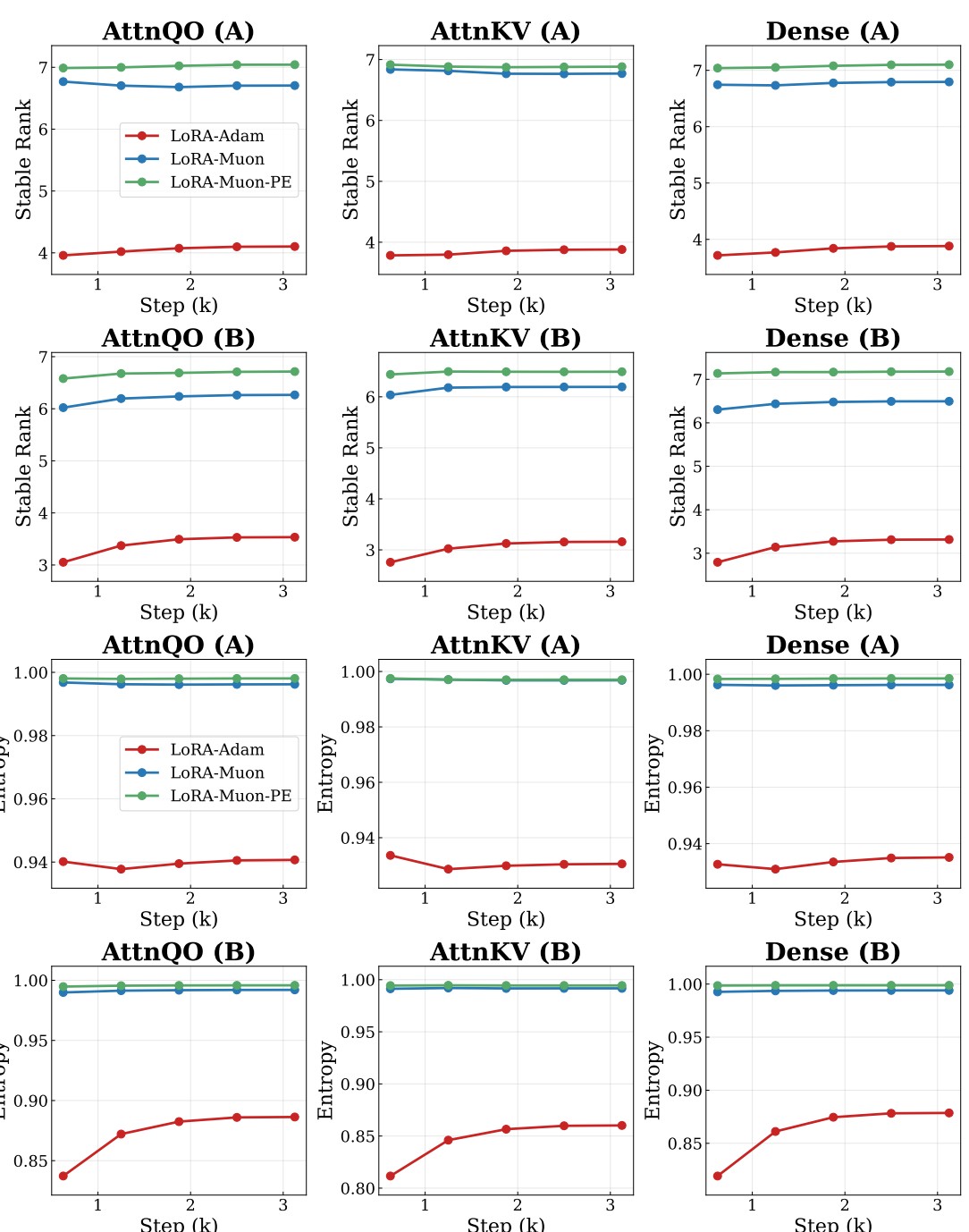

*Figure 12.* Stable rank and SVD entropy of LoRA matrices during Llama 2-7B fine-tuning on MetaMath.

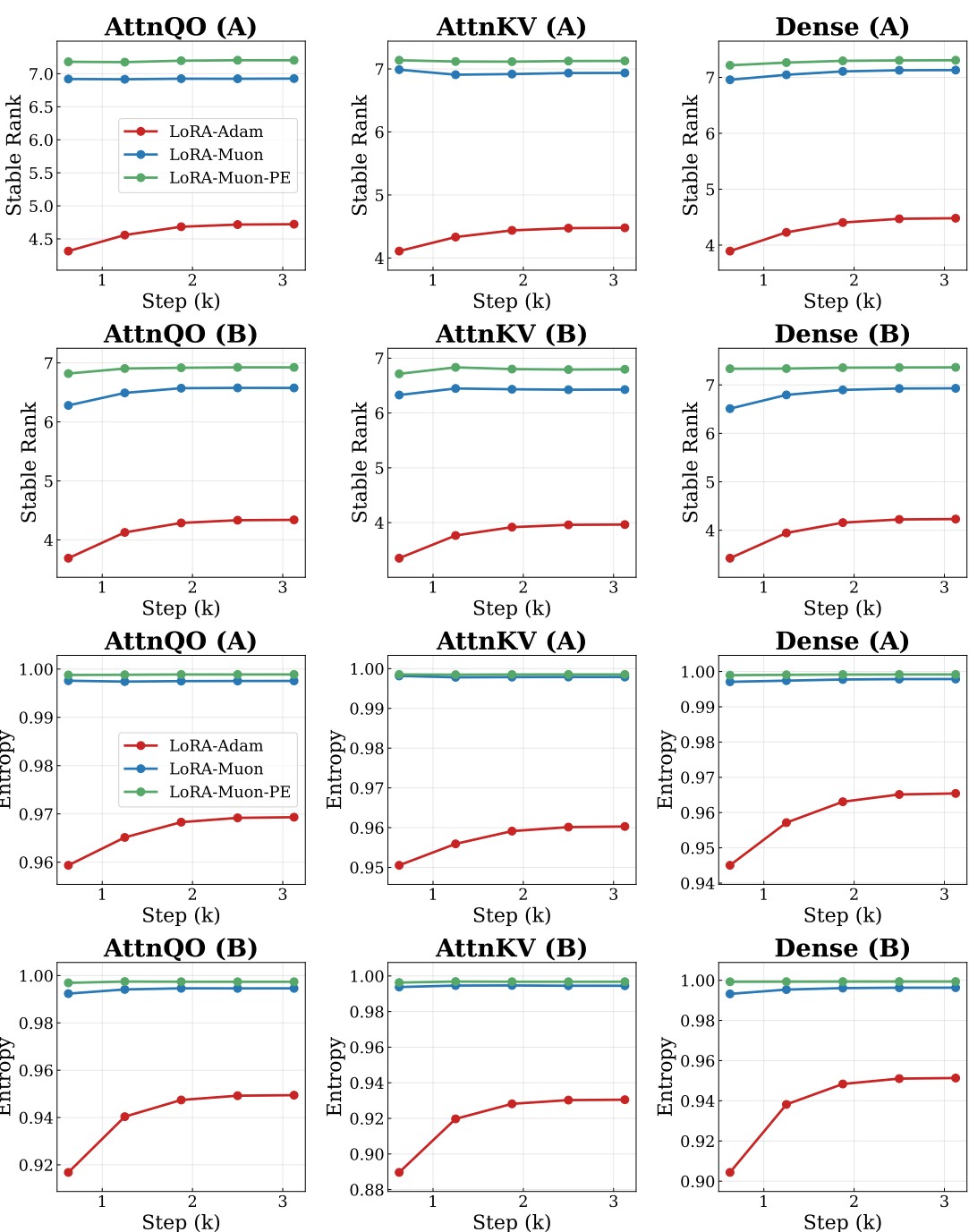

*Figure 13.* Stable rank and SVD entropy of LoRA matrices during Llama 2-7B fine-tuning on CodeFeedback.

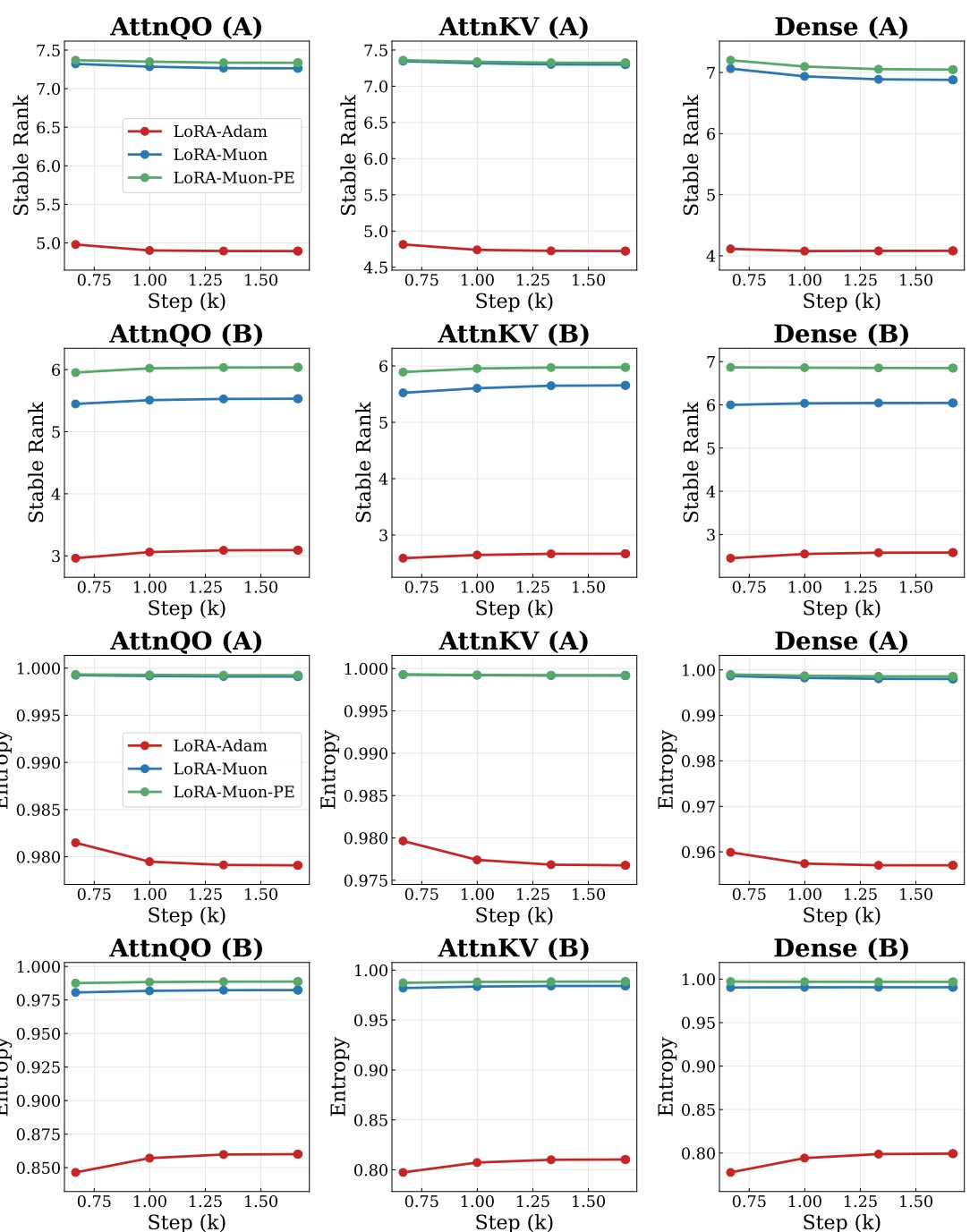

*Figure 14.* Stable rank and SVD entropy of LoRA matrices during Llama 2-7B fine-tuning on WizardLM (commonsense).

