# OpenReview forum: "Can Muon Fine-tune Adam-Pretrained Models?"
_ICML.cc/2026/Conference — ICML 2026 regular_

### Official Review · Reviewer_AHmn · 2026-03-09

**Soundness:** 4
**Presentation:** 4
**Significance:** 4
**Originality:** 4
**Overall Recommendation:** 5
**Confidence:** 4

**Summary:**

The authors in this paper study the optimizer mismatch problem that occurs when switching between Adam and Muon optimizers across pretraining and fine-tuning phases. Specifically, they investigate the degraded performance observed when Adam-pretrained models are fine-tuned with Muon, and vice versa. The authors attribute this mismatch to the distinct implicit biases of the optimizers: Adam is biased towards max-norm solutions via element-wise preconditioning, whereas Muon is biased towards spectral-norm solutions via matrix-level preconditioning. Through controlled experiments on both vision and language backbones, the authors show that full fine-tuning with a mismatched optimizer disrupts the structural knowledge established during pretraining. They also demonstrate that constraining the fine-tuning updates using Low-Rank Adaptation (LoRA) effectively mitigates this problem. Extensive evaluations show that LoRA-Muon can match or outperform LoRA-Adam across various tasks when fine-tuning Adam-pretrained models.

**Compliance With Llm Reviewing Policy:**

Affirmed.

**Final Justification:**

The rebuttal successfully addressed my main concerns regarding spectral dynamics of the LoRA matrices (A and B) during fine-tuning. This clarification changed my evaluation, and I am pleased to confirm my increased score.

**Key Questions For Authors:**

While you note that Muon-pretrained weights exhibit higher stable rank and SVD entropy compared to Adam, have you investigated the spectral dynamics of the LoRA matrices (A and B) themselves during fine-tuning? Analyzing this could provide insight into why LoRA effectively resolves the optimizer mismatch, occasionally even allowing it to surpass the performance of a matched pretraining and fine-tuning pipeline.

**Limitations:**

yes

**Strengths And Weaknesses:**

**Strengths**

Novelty: the paper presents the first in-depth analysis of the optimizer mismatch problem that occurs when switching between optimizers across pretraining and fine-tuning stages.

Analysis on Implicit Biases: the authors provide a compelling analysis linking the mismatch problem to the fundamentally different implicit biases of the two optimizers. Specifically, they effectively contrast Adam's element-wise preconditioning (biased toward max-norm) with Muon's matrix-level preconditioning (biased toward spectral-norm).

Robust Empirical Validation: the authors back their claims with strong empirical results across natural language and image classification tasks. Furthermore, they strengthen their case with controlled experiments, pretraining 561M-parameter NanoChat models from scratch with both Muon and Adam to cleanly isolate the mismatch phenomenon.

Relevant in practice: because most publicly available open-source models are pretrained with Adam, the optimizer mismatch severely limits the adoption of the efficient Muon optimizer for fine-tuning. By proving that standard LoRA effectively mitigates this limitation, the authors provide actionable utility for practitioners wanting to leverage Muon's capabilities.


**Weaknesses**


Simplifications in theoretical analysis: the theoretical analysis of the optimizers' implicit biases is restricted to a simplified linear regression setting. To make the optimization dynamics tractable, the analysis omits momentum and uses SignGD as a proxy for Adam. While this yields valuable intuition, a formal theoretical characterization of how mismatched implicit biases disrupt pretrained knowledge in deep neural networks remains an open question, which the authors transparently acknowledge.

---

> ### Author Rebuttal · Authors · 2026-03-30
>
> We thank the reviewer for the positive assessment and insightful question.
>
> ## On simplifications in theoretical analysis (Weakness)
>
> As the reviewer notes, our linear regression analysis is intended to provide accessible intuition for implicit bias differences, complementing our empirical investigation. While the setting is simplified, it already clearly reveals that the two optimizers converge to fundamentally different solution geometries. We then validate that this mismatch manifests at larger scales through extensive experiments on T5, Llama2-7B, and CLIP ViT-B/32. Concurrent work is also making progress on analyzing implicit biases in more complex settings, some of which we reference in Section 3.1. We are optimistic that as the theoretical understanding of Muon advances, our empirically validated insights can be further grounded in formal analysis.
>
> ## Spectral dynamics of LoRA matrices (A and B)
>
> This is an insightful suggestion. Following the reviewer's question, we analyzed the spectral properties of the LoRA A and B matrices separately during Llama2-7B fine-tuning across all three language tasks (Math, Code, Commonsense), using the same settings as Table 3. We measure stable rank and normalized SVD entropy throughout training for attention (Q/K/V/O) and dense layers. **The detailed figures are available at https://ibb.co/album/QpQGnT.**
>
> We observe that LoRA-Muon consistently produces a higher stable rank ($\sim$6--7 vs $\sim$3--5 for Adam) and entropy ($\sim$0.98--1.0 vs $\sim$0.80--0.95 for Adam) across all layer types, mirroring the patterns in pretrained weights (Section 3.1). This confirms that Muon's implicit bias toward uniform singular value distributions extends to the LoRA matrices.
>
> Notably, LoRA learns on freshly initialized A and B matrices rather than directly modifying pretrained weights. This may explain why LoRA mitigates mismatch: Muon can freely express its implicit bias (which may benefit generalization [1]) on these new matrices, while the pretrained knowledge remains preserved in the frozen base weights. In contrast, full fine-tuning forces Muon to directly alter Adam-shaped weights, causing disruption. Establishing a more rigorous explanation requires further investigation.
>
> ---
>
> We appreciate the reviewer's constructive feedback. We will incorporate the spectral analysis figures in the revised paper. We hope our responses have addressed the reviewer's concerns, and we would be grateful if the reviewer would consider updating their score accordingly. We are happy to address any further questions.
>
> [1] Vasudeva et al., "How Muon's Spectral Design Benefits Generalization: A Study on Imbalanced Data," arXiv 2025.

---

> > ### Author Rebuttal · Reviewer_AHmn · 2026-04-02
> >
> > Thank you for your response. It has clarified my concerns, and I confirm I am raising my score.

---

> > > ### Author Response · Authors · 2026-04-07
> > >
> > > We sincerely thank the reviewer for the positive feedback, for raising the score, and for the valuable time and effort devoted to reviewing our work. We will incorporate the spectral analysis in the revision.

---

### Official Review · Reviewer_NkEp · 2026-03-11

**Soundness:** 2
**Presentation:** 3
**Significance:** 2
**Originality:** 2
**Overall Recommendation:** 3
**Confidence:** 4

**Summary:**

The paper makes two main contributions: it offers the detailed study of the optimizer mismatch problem between Adam pretraining and Muon fine-tuning, linking the gap to differing implicit biases and the disruption of pretrained knowledge, and it shows that constraining updates with LoRA effectively mitigates this issue, allowing LoRA-Muon to match or outperform LoRA-Adam across both language and vision tasks.

**Compliance With Llm Reviewing Policy:**

Affirmed.

**Final Justification:**

The rebuttal is helpful in that it clarifies the paper’s intended contribution, adds a larger-scale Llama2-13B result showing LoRA-Muon remains competitive with LoRA-Adam, and better explains the observed rank sensitivity as evidence that constraining updates can mitigate optimizer mismatch at low-to-moderate ranks; however, I still find the overall case only partially resolved because the theory-to-method connection remains largely heuristic—LoRA is motivated by the mismatch analysis but not studied within the same theoretical framework—and the practical guidance is still limited, since the rebuttal stops short of giving clear actionable conditions for when one should prefer Adam vs. Muon or how to choose rank, learning rate, and constraint strength in practice.

**Key Questions For Authors:**

1. Can the authors provide experiments on larger-scale or more realistic Adam-pretrained backbones?

2. How to interpret the strong rank sensitivity of LoRA-Muon on MetaMath?

**Limitations:**

yes

**Strengths And Weaknesses:**

**Strengths**

This paper identifies the Adam-to-Muon fine-tuning mismatch, tries to explain it, and offers a simple, practical fix via LoRA, which is verified across both language and vision tasks and makes the contribution both insightful and useful in practice.



**Weaknesses**

1. **The empirical finding is not new.**

    The empirical observation of optimizer mismatch between pretraining and fine-tuning is not entirely new, as related phenomena have already been discussed as the author mention in section 3. In this paper, the main added value comes from controlled experiments and an analysis based on a toy linear regression setup. However, 1) the controlled study is still quite limited in scale and coverage, which weakens the novelty and impact of the paper. 2) while the paper uses a linear regression toy model to analyze the mismatch between AdamW and Muon, the proposed mitigation strategy, namely LoRA, is not grounded in a corresponding analysis on the same toy setup. As a result, the theoretical explanation and the practical solution are not fully connected.

2. **The practical guidance is not significant.**

    Although the paper compares different optimizer choices across pretraining and fine-tuning, it remains unclear what practitioners should actually take away from the results. In particular, the paper does not clearly answer when one should prefer Adam-style optimization, when Muon is safe to use, and under what conditions LoRA is a reliable remedy. This limits the practical usefulness of the work for real-world model development and deployment.

3. **The main conclusion seems highly task-dependent.**

    The rank-scaling results suggest that the effectiveness of LoRA-Muon depends strongly on the task regime. In MetaMath, where optimizer mismatch is the most pronounced, increasing LoRA rank causes LoRA-Muon to degrade and gradually approach full fine-tuning behavior, whereas this effect is much weaker on code and vision tasks. This indicates that the proposed solution may not be robust across domains, and the current presentation somewhat overstates the generality of the conclusion.

---

> ### Author Rebuttal · Authors · 2026-03-30
>
> We thank the reviewer for the detailed feedback and address each point below.
> ## W1: The empirical finding is not new
>
> We do not claim the observation of optimizer mismatch as novel—indeed, we explicitly cite prior work noting this phenomenon. Our contribution lies in providing the first systematic investigation into why this mismatch occurs and how it can be mitigated. The core insight is that Adam and Muon induce fundamentally different implicit biases, and switching optimizers disrupts pretrained knowledge; we hypothesize that constraining updates should mitigate this. We validate this through novel empirical findings—LoRA fine-tuning, rank-scaling, learning rate analysis, forgetting measurements, and LoRA-variant comparisons.
>
> - **Scale.** Pretraining is expensive, and we carefully tuned hyperparameters separately for Adam and Muon to ensure reliable comparison. Given resource constraints, we leave larger-scale controlled studies to future work, as acknowledged in our limitations.
>
> - **Theory-LoRA connection.** While our theoretical analysis is based on simplified linear regression, this already reveals that the two optimizers converge to fundamentally different solution geometries. We then validate that this mismatch manifests at larger scales through extensive experiments on T5, Llama2-7B, and CLIP ViT-B/32. As presented in our paper, our insight on implicit biases (Section 3.1), combined with the optimal learning rate analysis (Figure 4), leads us to hypothesize that constraining updates should mitigate the mismatch (Section 3.2); we then choose LoRA to test this hypothesis as it is the most widely-adopted method for constraining updates.
>
> ## W2: The practical guidance is not significant
>
> Our core insight that constraining updates mitigates optimizer mismatch directly informs fine-tuning strategy, and can inspire approaches beyond those explored in this paper, such as adding regularization, as mentioned in our Conclusions. In the settings we study, we validate that simply applying LoRA is already an effective option, allowing Muon to match Adam's performance while providing a more memory-efficient alternative (50\% reduction in optimizer states). This offers a ready-to-use starting point for leveraging Muon's efficiency. Given that Muon research is at an early stage and ours is the first systematic study of Muon fine-tuning, we do not claim the final solution; rather, we establish the practical viability of using Muon for fine-tuning Adam-pretrained models, and future work will build on these findings.
>
> ## W3: The main conclusion seems highly task-dependent
>
> Our hypothesis that constraining updates mitigates optimizer mismatch is consistently validated across all tasks. On language tasks (Figure 4), LoRA-Muon outperforms LoRA-Adam at moderate ranks where updates are constrained while retaining sufficient capacity; at higher ranks, behavior approaches full fine-tuning, and LoRA-Muon begins to underperform, exactly as our hypothesis predicts. On vision tasks (Figure 5), the mismatch is minor, and LoRA-Muon outperforms LoRA-Adam on average across ranks. Beyond validating our hypothesis, by extending prior work [1] to more settings, we find that mismatch severity varies across scenarios. This aligns with evidence that fine-tuning is task-dependent [2], offering a more complete picture for future research.
>
> ## Q1: Larger scale experiments
>
> We extended our LoRA experiments to Llama2-13B on CodeFeedback using the same settings as Table 3. We swept learning rates in [1e-5, 9e-4] and report results averaged over 3 seeds (best LR = 7e-4 for both):
>
> - LoRA-Adam: 33.17\% ($\pm$1.17\%)
> - LoRA-Muon: 34.76\% ($\pm$2.44\%)
>
> LoRA-Muon performs comparably to LoRA-Adam at this larger scale, consistent with Llama2-7B, providing evidence that our insight generalizes. For full fine-tuning at 13B, memory constraints prevent standard DDP (which we use to ensure fair comparison with the original Muon implementation); we leave this to future work as Muon's compatibility with memory-reduction frameworks improves.
>
> ## Q2: Rank sensitivity on MetaMath
>
> We would like to clarify that we explicitly use the rank sensitivity on MetaMath to support our hypothesis in Section 4.4. At low to moderate ranks, LoRA constrains updates, allowing LoRA-Muon to match or outperform LoRA-Adam, with best results at moderate ranks balancing constraint and expressiveness. At higher ranks, the constraint relaxes and behavior approaches full fine-tuning, where mismatch exists on MetaMath. This causes LoRA-Muon to degrade while LoRA-Adam continues to improve. This rank sensitivity reflects how constraining updates mitigates mismatch, supporting our hypothesis.
>
> ---
> We will incorporate these clarifications and hope this addresses the reviewer's concerns.
>
> [1] Liu et al., "Muon is Scalable for LLM Training," arXiv 2025.
>
> [2] Zhang et al., "When Scaling Meets LLM Finetuning: The Effect of Data, Model and Finetuning Method," ICLR 2024.

---

> > ### Author Rebuttal · Reviewer_NkEp · 2026-04-02
> >
> > Thank you for the rebuttal. While the additional clarifications are helpful, I believe two core concerns remain insufficiently addressed.
> >
> > (1) Weak theory–method connection.
> > The rebuttal reiterates that the theoretical analysis reveals a geometry mismatch between optimizers and that this insight motivates the use of LoRA to constrain updates. However, the connection remains largely heuristic. In particular, the proposed mitigation (LoRA) is not analyzed within the same theoretical framework used to study the mismatch (i.e., the toy linear regression setup). As a result, it is still unclear whether LoRA can provably mitigate the mismatch under the same assumptions, or whether the observed improvements are purely empirical. Strengthening this connection—e.g., by demonstrating that LoRA-style constraints reduce mismatch in the controlled setting—would significantly improve the conceptual rigor of the paper.
> >
> > (2) Limited practical guidance.
> > The response to W2 mainly reframes the contribution as providing a “starting point” rather than concrete guidance. While this clarification is appreciated, it effectively weakens the original claim without resolving the concern. The paper still falls short of offering actionable insights for practitioners. For example, it remains unclear under what conditions one should prefer Adam vs. Muon, when LoRA is a reliable remedy, and how key factors (e.g., rank, learning rate, or degree of constraint) should be chosen in practice. Without such guidance, the practical impact of the work is limited.
> >
> > Overall, based on the rebuttal, I still keep my score.

---

> > > ### Author Response · Authors · 2026-04-07
> > >
> > > We thank the reviewer for the continued discussion. Below, we address the two concerns raised.
> > >
> > > ## On the theory--method connection
> > >
> > > We extend the one-sample linear-regression framework from Appendix D to cover both full fine-tuning and LoRA, using SignGD as a tractable proxy for Adam. Due to space limits, we sketch only the *SignGD-pretrained* case here; the Muon-pretrained case is analogous, and the complete proof will appear in the revision.
> > >
> > > **Setup.** Let $W_0$ be the pretrained weight obtained via SignGD, and let $(z,b)$ be a fine-tuning sample with residual $r_0 := b - W_0 z$. The fine-tuning loss over the weight update $\Delta := W - W_0$ is $$L_{\mathrm{ft}}(\Delta) = \frac{1}{2}\Vert\Delta z - r_0\Vert_2^2.$$
> > >
> > > **Fine-tuning.** To align with SignGD's pretraining geometry, we consider updates satisfying $\Vert\Delta\Vert_{\max} \le \rho$: $$\min_{\Vert\Delta\Vert_{\max}\le \rho} \frac{1}{2}\Vert\Delta z-r_0\Vert_2^2.$$
> > > The minimum norm level achieving zero loss is $$\rho_{\mathrm{s}}^\star = \Vert r_0\Vert_\infty / \Vert z\Vert_1.$$ By contrast, the zero-loss update selected by Muon is $$\Delta_{\mathrm{m}}^\star = r_0 z^\top / \Vert z\Vert_2^2,$$ which requires $$\Vert\Delta_{\mathrm{m}}^\star\Vert_{\max} = \Vert r_0\Vert_\infty \Vert z\Vert_\infty / \Vert z\Vert_2^2 \ge \rho_{\mathrm{s}}^\star.$$ Hence, under the same max-norm budget, there exists a range of norm levels where SignGD achieves zero loss while Muon does not, formalizing how optimizer mismatch harms fine-tuning.
> > >
> > > **Forgetting.** For a pretrained sample satisfying $W_0 x = y$, any update $\Delta$ incurs $$L_{\mathrm{old}}(\Delta) = \frac{1}{2}\Vert\Delta x\Vert_2^2 \le \frac{m}{2}\Vert\Delta\Vert_{\max}^2 \Vert x\Vert_1^2.$$ Thus, in the SignGD-pretrained case, the zero-loss SignGD update yields a tighter max-norm-based forgetting bound than the zero-loss Muon update.
> > >
> > > **LoRA.** In this toy model, full fine-tuning updates are already rank-one, so standard LoRA is too expressive. To isolate the effect of low-rank constraints, we consider a fixed-subspace surrogate $W = W_0 + BA$, where $A \in \mathbb{R}^{r \times n}$ is fixed and only $B$ is trained. The loss becomes $$L(B) = \frac{1}{2}\Vert BAz - r_0\Vert_2^2,$$ so the same analysis applies with $z$ replaced by $Az$.
> > >
> > > The SignGD-aligned threshold becomes $$\rho_{\mathrm{s,LoRA}}^\star = \Vert r_0\Vert_\infty / \Vert Az\Vert_1,$$ while Muon requires $$\tilde\rho_{\mathrm{m,LoRA}}^\star = \Vert r_0\Vert_\infty \Vert Az\Vert_\infty / \Vert Az\Vert_2^2.$$ The mismatch inflation factor satisfies $$\tilde\rho_{\mathrm{m,LoRA}}^\star / \rho_{\mathrm{s,LoRA}}^\star \le \Vert Az\Vert_0 \le r,$$ where $\Vert\cdot\Vert_0$ counts nonzero entries. This yields an explicit rank-dependent control: lower-rank adapters impose a tighter worst-case upper bound on mismatch inflation. At $r = 1$, the two thresholds coincide, and the mismatch vanishes; when $A = I$, we recover full fine-tuning. In practice, however, very small ranks may underfit complex tasks, which explains why moderate ranks often perform best ---they balance mismatch mitigation against expressiveness.
> > >
> > > ## On practical guidance
> > >
> > > We'd like to clarify that our experiments already contain actionable guidance for fine-tuning Adam-pretrained models with Muon. We summarize them explicitly here and will add this summary to the revision:
> > >
> > > 1. **Use LoRA** (Sections 3.2--4.3). Compared to full fine-tuning, LoRA reduces the gap between Muon and Adam: LoRA-Muon matches LoRA-Adam across our benchmarks. Since Muon has 50\% lower optimizer-state memory than Adam, LoRA-Muon can serve as a drop-in replacement for LoRA-Adam when memory is constrained.
> > >
> > > 2. **Tune learning rate separately** (Figure 4 and all LR sweeps). The optimal LR for Muon often differs from Adam's, especially under full fine-tuning. Directly reusing Adam's LR for Muon typically underperforms; separate tuning is necessary.
> > >
> > > 3. **Use moderate ranks** (Sections 4.4--4.5). The optimal rank is neither too small nor too large: very small ranks underfit complex tasks, while very large ranks approach full fine-tuning and reintroduce mismatch. The precise sweet spot is task-dependent, but moderate ranks generally balance learning capacity against mismatch severity.
> > >
> > > 4. **Do not directly transfer LoRA variants** (Section 4.6). LoRA variants benchmarked and optimized with LoRA-Adam do not necessarily transfer to LoRA-Muon; their relative performance may differ.
> > >
> > > We regard this work as a starting point toward making Muon fine-tuning as general and powerful as Adam fine-tuning. We believe that better methods for using LoRA, or approaches that enable full-Muon to match full-Adam, remain to be discovered, as stated in our Limitations section. Through such methods, we hope to better transfer Muon's demonstrated pretraining advantages to the fine-tuning setting.
> > >
> > > ---
> > >
> > > We hope these clarifications address the reviewer's concerns. We are happy to discuss further.

---

### Official Review · Reviewer_PcGi · 2026-03-13

**Soundness:** 3
**Presentation:** 3
**Significance:** 3
**Originality:** 3
**Overall Recommendation:** 4
**Confidence:** 4

**Summary:**

See below

**Compliance With Llm Reviewing Policy:**

Affirmed.

**Final Justification:**

I have carefully read the rebuttal by authors. I will keep my score and vote for acceptance.

**Key Questions For Authors:**

**Main Contributions:**

The paper first identifies and studies the "optimizer mismatch" problem when fine-tuning Adam pretrainedmodels using the Muon optimizer. It then finds that LoRA can mitigate this mismatch. Experiments are conducted across language and vision tasks.  the authors also provide an initial theoretical analysis from the perspective of implicit biases.



**Strengths:**

The presentation is mostly clear, and the figures are visually enjoyable to read. The problem is practically important, as most publicly available models are pretrained with Adam.  The paper provides a clear hypothesis (optimizer mismatch disrupts pretrained knowledge) and tests it through multiple lenses. The empirical evaluation is thorough, covering multiple tasks (e.g., NLU, NLG, image classification) and model scales (e.g., T5-Base, Llama 2-7B, CLIP ViT-B/32).



Overally speaking, this paper is well-written and  well-executed. I think the conclusion in the paper is worth sharing with the community.





**Weaknesses (and my major concerns):**

The experimental results, while consistent, show relatively small absolute differences in many cases (e.g., Table 2, Table 3). The paper argues this is because methods are well-tuned and near convergence, which is reasonable. However, it raises the question of practical significance. For practitioners, is the benefit of switching to LoRA-Muon worth the engineering effort, given that LoRA-Adam already performs well?

**Limitations:**

See above

**Strengths And Weaknesses:**

See below

---

> ### Author Rebuttal · Authors · 2026-03-30
>
> We thank the reviewer for the positive assessment and for raising this meaningful discussion. We appreciate the opportunity to clarify.
>
> ## Regarding practical significance
>
> **On the small absolute differences.** Small performance gaps across methods are common on well-tuned fine-tuning benchmarks. A recent study [1] similarly finds that when different LoRA methods are tuned to their optimal learning rates, they yield comparable performance, with gaps of a similar magnitude to those we report. This is a general phenomenon in mature benchmarks rather than a limitation specific to Muon. Since there is no free lunch, and Muon is fundamentally different from Adam-type optimizers, we believe that once those challenges around Muon are better addressed, Muon fine-tuning can offer superior performance in certain practical scenarios.
>
> **On the main takeaway of Section 4.** We believe the key contribution lies not in absolute accuracy gains, but in empirically validating our hypothesis from Section 3: constraining update magnitude mitigates optimizer mismatch. By simply applying LoRA to limit updates, Muon transforms from underperforming Adam to showing potential to outperform it (Tables 2-3). The rank-scaling and LoRA-variant experiments further confirm that as the update magnitude increases, this mitigation effect diminishes. As an early-stage research direction, we do not claim that using LoRA, especially in our relatively simple manner, is the best practice or final solution for fine-tuning with Muon. As discussed in our Conclusions, we believe there exist other approaches originated from our insights to address the mismatch issue, such as aligning initializations before fine-tuning. As a first step in this direction, we hope our work and the insights it provides can inspire further research, ultimately leading to better solutions.
>
> **Practical benefits of Muon.** Beyond accuracy, Muon offers concrete advantages for fine-tuning: (1) 50\% reduction in optimizer states, significantly lowering memory; (2) faster learning in the early stage of training, achieving good performance with less data [2]. These benefits are particularly valuable for resource-constrained scenarios, such as fine-tuning on consumer hardware or on-device learning for model personalization.
>
> **On engineering effort.** We acknowledge Muon's ecosystem is still in its early stages, both in research and practice. However, as more research emerges to better understand Muon and address its current limitations, we expect growing community adoption and integration into mainstream open-source frameworks, thereby reducing practical barriers over time.
>
> ---
>
> We appreciate the reviewer's positive assessment and valuable discussion. We will incorporate the clarifications into the revised paper. We hope our responses have addressed the reviewer's concerns, and we would be grateful if the reviewer would consider raising their score. We are happy to address any further questions.
>
> [1] Lee et al., "Learning Rate Matters: Vanilla LoRA May Suffice for LLM Fine-tuning," arXiv 2026.
>
> [2] Liu et al., "Muon is Scalable for LLM Training," arXiv 2025.

---

> > ### Author Rebuttal · Reviewer_PcGi · 2026-04-02
> >
> > I would like to thank the authors for the detailed rebuttal. I will keep my score and vote for acceptance.

---

> > > ### Author Response · Authors · 2026-04-07
> > >
> > > We sincerely thank the reviewer for the positive feedback and support, as well as for the time and effort in reviewing our work. We will incorporate the suggested clarifications in the revision.

---

### Official Review · Reviewer_JFpr · 2026-03-13

**Soundness:** 2
**Presentation:** 3
**Significance:** 2
**Originality:** 2
**Overall Recommendation:** 4
**Confidence:** 3

**Summary:**

The paper studies whether Muon can be used to fine-tune models pretrained with Adam-like adaptive optimizers. It first reproduces a pretraining/fine-tuning optimizer-mismatch effect in a controlled NanoChat setup, argues that Adam and Muon induce different structural biases in pretrained weights, and then tests the hypothesis that restricting updates with LoRA reduces the resulting degradation. The experiments span language and vision benchmarks, LoRA-rank sweeps, catastrophic-forgetting measurements, and comparisons to several LoRA variants.

**Compliance With Llm Reviewing Policy:**

Affirmed.

**Final Justification:**

The rebuttal addresses my main concerns well enough to move me up from weak reject. The most important improvement is the added statistical analysis. The pooled meta-analysis is a better test of the paper’s actual claim than focusing on tiny raw average differences, because the claim is that LoRA reduces optimizer mismatch. I still do not think the generic regularization and smaller effective updates are not completely disentangled, but I now think the evidence is sufficient for the paper’s scope. I also still think the title is slightly broader than what is demonstrated, and the practical fine-tuning case remains secondary even after the added wall-clock/memory discussion.

**Key Questions For Authors:**

Please provide per-task mean and std and significance tests for the aggregate numbers in Tables 2–4.

What direct evidence shows that LoRA helps because it preserves pretrained structure, rather than because it is simply a generic regularizer or an effective reduction in update magnitude?

Why was LoFT not included in the LoRA-variant study, given that the paper already cites work aimed at making low-rank adaptation closer to full fine-tuning and Table 6 is presented as a broader test of LoRA variants?

**Limitations:**

Matched vs mismatched pretraining is only done on NanoChat at 561M parameters, the NLU model is Adafactor-pretrained T5, and the largest Adam-pretrained language model studied is Llama 2-7B.

**Strengths And Weaknesses:**

Strengths
The paper makes a useful step beyond the prior observation of Muon/AdamW mismatch by proposing a concrete mitigation hypothesis, and testing it across controlled, language, and vision settings.

The controlled NanoChat study motivates the later experiments, the toy analysis in Section 3 is clearly separated from the larger empirical claims, and the appendix gives concrete optimizer settings, learning-rate choices, and hardware details, so the work is reproducible.

Weaknesses
The title is broader than the demonstrated result, since the paper mostly shows that LoRA-Muon can roughly match LoRA-Adam, not that full Muon fine-tuning on Adam-pretrained models is competitive. The practical case is incomplete because no wall-clock, throughput, or memory results are reported for fine-tuning.

Many reported gains are very small, especially on GLUE and CLIP averages, yet no standard deviations or confidence intervals are shown.

---

> ### Author Rebuttal · Authors · 2026-03-30
>
> We thank the reviewer for the thoughtful feedback and address each point below.
>
> ## On standard deviations and significance (Q1 and Weakness)
>
> Small absolute differences are common on well-tuned fine-tuning benchmarks—[1] finds that different LoRA methods yield comparable performance when properly tuned. More importantly, our core claims do not rest on these absolute gaps. Rather, the experiments in Section 4 are designed to validate our insight from Section 3: that constraining update magnitude mitigates optimizer mismatch.
>
> Tables 2--4 in the submission already report mean performance over 3 seeds. **Due to space limits, we provide the updated tables at https://ibb.co/1tV6FHNT.** We performed a significance analysis targeted at our main claim. We compute the reduction in Adam--Muon gap when switching from full fine-tuning to LoRA, then aggregate across tasks using random-effects meta-analysis. Across Tables 2--4, the pooled effect is positive for both Muon and Muon-PE: 0.72\% (95\% CI: [0.41, 1.04], two-sided $p<0.001$) and 0.83\% (95\% CI: [0.45, 1.20], two-sided $p<0.001$), respectively. This supports our claim that LoRA mitigates the Adam--Muon mismatch across tasks, while allowing heterogeneity across benchmarks.
>
> ## On the scope of our paper (Weakness about title)
>
> The title poses an open question with no prior systematic study—only [2] noted Muon underperforms on Adam-pretrained models. We provide the first in-depth investigation into why and how to address this. Our central contribution is the insight that mismatch degrades performance by disrupting pretrained knowledge, and that *constraining update magnitude mitigates this issue*. The LoRA experiments in Section 4 validate this hypothesis from several angles:
>
> - By simply applying LoRA to limit updates, Muon transforms from underperforming to competitive across language and vision benchmarks.
> - Rank-scaling (Section 4.4), forgetting (Section 4.5), and LoRA variants (Section 4.6) further confirm that as update magnitude increases, mitigation shrinks.
>
> These experiments validate our hypothesis, rather than proposing LoRA—especially in our relatively simple manner—as the final solution. As discussed in Conclusions, better LoRA usage or other approaches (e.g., aligning initializations) could further address mismatch. We hope our work inspires further research on transferring Muon's benefits to fine-tuning.
>
> ## On wall-clock, throughput, and memory results
>
> Our paper focuses on understanding and mitigating the optimizer mismatch problem, rather than optimizing for practical efficiency. That said, we provide some context:
>   - *On training time.* Muon's ecosystem is still early compared to Adam's heavily optimized frameworks. While Muon's Newton-Schulz iteration adds overhead, [2] finds Muon is $\sim$2× more efficient under compute-optimal training. **We provide detailed wall-clock comparisons at https://ibb.co/album/Pcm1Dk.** For LoRA, LoRA-Muon is only 1.1--1.2$\times$ slower on Llama2-7B and 1.0--1.2$\times$ on CLIP. For full fine-tuning on Llama, Full-Adam requires ZeRO-2 while Full-Muon's memory savings enable DDP; this accounts for the larger gap (2.3--2.9$\times$). On CLIP, Full-Muon is only 1.0--1.2$\times$ slower.
>   - *On memory.* Muon requires only momentum (1×) vs Adam's momentum + second moment (2×), saving 50% optimizer states. For Llama 2-7B, this saves $\sim$14GB in FP32; for CLIP, $\sim$335MB.
> ## Q2: Direct evidence that LoRA preserves pretrained structure
>
> In Section 4.5, we measure pretrained knowledge disruption via catastrophic forgetting, showing LoRA-Muon exhibits less forgetting than Full-Muon. As suggested, we also measured the distance between fine-tuned and pretrained weights for the runs in Table 3, using L2 and cosine distance. **Due to space limits, we present the full table at https://ibb.co/album/smPHhb.** Under full fine-tuning, Muon's cosine distance is 5.6--7.4$\times$ larger than Adam on Math/Code, confirming more aggressive updates. Under LoRA, this reverses: Muon's cosine distance is only 0.2--0.8$\times$ of Adam's. On Commonsense where Full-Muon outperforms Full-Adam, Muon's distance is already smaller (0.65--0.75$\times$), and LoRA further reduces it to 0.15--0.18$\times$. Together, these provide direct evidence that LoRA preserves pretrained knowledge.
>
> ## Q3: Why LoFT was not included
>
> We experimented with LoFT, but after tuning, it did not outperform LoRA-Adam on our benchmarks. For reference, on GLUE with T5-Base, LoFT achieves an average of 88.83\% (CoLA: 82.45±0.75\%, MNLI: 86.14±0.07\%, MRPC: 87.99±0.42\%, QNLI: 93.15±0.11\%, SST-2: 94.42±0.24\%), vs. LoRA-Adam's 88.93\%. Among similar methods, we included LoRA-Pro and LoRA-RITE, which outperformed LoRA-Adam.
>
> ---
> We will incorporate these results and hope this addresses the reviewer's concerns.
>
> [1] Lee et al., "Learning Rate Matters: Vanilla LoRA May Suffice for LLM Fine-tuning," arXiv 2026.
>
> [2] Liu et al., "Muon is Scalable for LLM Training," arXiv 2025.

---

### Decision · Program_Chairs · 2026-04-30

**Decision:**

Accept (regular)

**Comment:**

The paper studies why Muon underperforms when fine-tuning Adam-pretrained models, attributes it to the optimizers' differing implicit biases, and shows that constraining updates with LoRA is an effective mitigation. Reviewers agreed the problem is timely and the study is careful and reproducible. Their concerns centred on the statistical support for small effects, the gap between the toy analysis and the proposed fix, and framing that is somewhat broader than what is demonstrated.

The rebuttal addressed these well. The added statistical analysis shows the effect is significant, and the additional theory and experiments connect the analysis to the proposed fix more directly; after discussion three of four reviewers support acceptance. The remaining reservation is that the connection is still partly heuristic and that the paper stops short of telling a practitioner when to prefer Muon over Adam or how to choose the LoRA rank for a given task. That is a fair description of the paper's limits but not of its correctness, and I view it as scope for follow-up work rather than a flaw in what is claimed. I recommend acceptance.

For camera-ready, please incorporate the rebuttal material and tighten the title and abstract to match the demonstrated result.

Congratulations!